# Identifying neural substrates of competitive interactions and sequence transitions during mechanosensory responses in *Drosophila*

**Jean-Baptiste Masson** [1,2]*, **François Laurent** [2], **Albert Cardona** [1,3,4], **Chloé Barré** [2], **Nicolas Skatchkovsky** [2], **Marta Zlatic** [1,4,5]*, **Tihana Jovanic** [1,2,6]*

**1** Janelia Research Campus, Howard Hughes Medical Institute, Ashburn, Virginia, United States of America, **2** Decision and Bayesian Computation, USR 3756 (C3BI/DBC) & Neuroscience Department, Institut Pasteur & CNRS, Paris, France, **3** Department of Physiology, Development, and Neuroscience, Cambridge University, Cambridge, United Kingdom, **4** MRC Laboratory of Molecular Biology, Trumpington, Cambridge, United Kingdom, **5** Department of Zoology, Cambridge University, Cambridge, United Kingdom, **6** Université Paris-Saclay, CNRS, Institut des Neurosciences Paris Saclay, Gif-sur-Yvette, France

* jean-baptiste.masson@pasteur.fr (J-BM); zlaticm@janelia.hhmi.org (MZ); tihana.jovanic@cnrs.fr (TJ)

**Data Availability Statement:** All relevant data are within the manuscript and its Supporting Information files.

## Abstract

Nervous systems have the ability to select appropriate actions and action sequences in response to sensory cues. The circuit mechanisms by which nervous systems achieve choice, stability and transitions between behaviors are still incompletely understood. To identify neurons and brain areas involved in controlling these processes, we combined a large-scale neuronal inactivation screen with automated action detection in response to a mechanosensory cue in *Drosophila* larva. We analyzed behaviors from $2.9 \times 10^5$ larvae and identified 66 candidate lines for mechanosensory responses out of which 25 for competitive interactions between actions. We further characterize in detail the neurons in these lines and analyzed their connectivity using electron microscopy. We found the neurons in the mechanosensory network are located in different regions of the nervous system consistent with a distributed model of sensorimotor decision-making. These findings provide the basis for understanding how selection and transition between behaviors are controlled by the nervous system.

## Author summary

All animals are constantly confronted with multiple behavioral options that they need to choose from in response to various stimuli. Yet how the brain controls the sensorimotor decision-making process and ensures that appropriate actions are chosen and correctly ordered remains a mystery. This is largely due to the difficulty in establishing causal relationships between neurons and sensorimotor decisions and determining the connectivity between the neurons underlying sensorimotor decisions. In this study, we used automated action detection to analyze behaviors of hundreds of thousands of *Drosophila* larvae in which we systematically silenced small subsets of neurons in response to a mechanosensory cue and identified candidate neurons for competitive interactions and transitions

**Funding:** We thank HHMI Janelia Research Campus for funding (M.Z., A.C., J-B.M and T.J.). We acknowledge funding from the Wellcome Trust Investigator Award 205050/Z/16/Z (M.Z) and Wellcome Trust International Recruitment Supplement 205050/A/16/Z (M.Z), from Institut Pasteur (J.-B.M), the sponsorship of CRPCEN & Gilead Science (J.-B.M), the ANR-17-CE23-0016 TRamWAy (J.-B.M), the "programme d'investissement d'avenir" supported by the "agence nationale de la recherche" ANR-19-P3IA-0001 (PRAIRIE 3IA Institute) (J.-B. M.), the INCEPTION project (PIA/ANR- 16-CONV-0005, OG) (J.-B.M), ANR-17-CE37-0019-01 (T.J.). This project has received funding from the European Union's Horizon 2020 research and innovation programme under the Marie Sklodowska-Curie grant agreement No 798050 (T.J.) The funders had no role in study design, data collection and analysis, decision to publish, or preparation of the manuscript.

**Competing interests:** The authors have declared that no competing interests exist.

between different actions. We further analyzed the connectivity between these neurons using electron microscopy reconstruction and identified a putative network underlying sensorimotor decision-making during mechanosensory responses. We found this mechanosensory network is distributed in the nervous system from the first order inter-neuron on the sensory side to the motor side and higher order brain regions.

## Introduction

In response to a sensory cue, animals can perform different behaviors. The transformation of the sensory information into appropriate motor outputs involves different parts of the nervous system from sensory processing areas to motor control areas. In addition, in order to enable reliable and coherent responses of organisms to sensory cues, the choice of one action must be accompanied by a full suppression of all competing physically mutually exclusive actions, thus areas of the nervous system that ensure proper selection of behavior will also be involved. Finally, animals often respond to stimuli, not with single actions but with sequences of actions in which case, the transitions between actions need to be precisely controlled.

In response to a single stimulus, different individuals may perform different actions and also the same individual can respond to the repetition of the stimulus differently. Thus, these responses can be probabilistic.

The circuit implementation of competitive interactions between neurons that promote mutually exclusive actions and the circuit mechanisms that control action stability and transitions from one action to the next are still incompletely understood.

For example, whether the choice of an action, its stability and transition into different actions are implemented in specialized centers that would implement competitive interactions between different option or distributed across the nervous system form the sensory to the motor side is still a heavily debated subject [1–9]. We have previously identified a circuit for behavioral choice (between two actions) and a two-element sequence in response to an air-puff at the earliest stage sensory processing site which supports a model in which competitive interactions are distributed across the nervous system [3,10]. To examine the generalization of this computing architecture, it is necessary to identify other sites of competitive interactions during sensorimotor decisions and understand how the outcomes of these putative circuits for behavioral choice are integrated to give rise to a unified choice about which action to perform.

In addition, when actions are executed one after the other in sequences, how do nervous systems specify the order of the individual actions in the sequence? There have been different theoretical models of sequence generation proposed: parallel queuing [11,12], synaptic chains [13–15], ramp-to-treshold [16–18] and more recently chains of disinhibitory loops [10] but how flexible behavioral sequences are implemented in nervous systems remains an open question.

A first step towards understanding the circuit mechanisms of the selection and transition between behaviors is to identify neurons and nervous system areas involved in promoting, stabilizing and suppressing specific actions.

Here, we used the GAL4/UAS system available in the *Drosophila* to selectively and constitutively silence neurons in populations of behaving *Drosophila* larvae during behavioral responses to an air-puff stimulation using tetanus-toxin. The air-puff can evoke five different actions in the larva that can be organized in various sequences in a probabilistic manner which makes it suitable for studying both competitive interactions and transitions between behaviors. We have previously used this assay to study the circuit mechanisms underlying the choice

between the two most prominent actions that occur in response to air-puff and a sequence of these two behaviors [10]. Here we expanded the study on multiple behaviors (of up to five different actions). We designed a behavioral screen using a library of *Drosophila* lines [19,20]. Based on anatomical pre-screen of a collection of images of the larval stages of the GAL4 *Drosophila* lines we selected those that target small subset of neurons. 567 driver lines were screened and the behaviors of hundreds of thousands of animals recorded. To quantify larval responses, we used supervised machine learning to reliably detect the actions that can be evoked by air puff. In addition to new sensory lines, we identified candidate GAL4 lines that target central neurons required for behavioral responses to mechanosensory stimuli. Amongst these, we identified candidate elements of functional circuit modules for mechanosensory responses (labeled by 66 GAL4 lines) out of which 25 GAL4 lines label candidate neurons for competitive interactions some of which are also involved in sequence transitions. Some of these are lines with sparse neuronal expression patterns so we characterized them in detail using single-cell FLP-out and electron microscopy reconstruction in a volume of the larval central nervous system [21,22] to determine their connectivity patterns. The candidate hit lines span different regions of the ventral nerve cord and the brain which suggests multiple sites of competition that are distributed across this nervous system.

These candidate lines provide a valuable resource for studying neural mechanisms underlying sensorimotor behaviors as it can restrict the focus for those interested in specific aspect of behavior (i.e. sensory processing, decisions, motor control, sequence generation) from the exploration of all the neurons in this nervous system to specific neurons and brain regions. Therefore this database is a prerequisite for the comprehensive circuit studies underlying sensorimotor decisions and action sequences during behavioral responses to a mechanical sensory cues as the candidate neurons can be used as starting points for circuit mapping and analysis by combining the results of silencing behavioral genetics experiments with the different methods available in *Drosophila* larva to relate structure and function of neuronal circuits: EM reconstruction, optogenetics, electrophysiology, calcium imaging, immunohistochemistry and modeling [10,21,23–30].

## Results

### An assay for sensorimotor decisions and sequences, using machine-learning based characterization of the behavioral response to an air-puff

In order to functionally map neurons specifically to the different aspects of sensorimotor behaviors, we presented stimuli to populations of freely behaving wild type larvae and larvae with inactivated neurons and monitored their behavioral responses (Fig 1A and 1B) to detect changes in behavior upon neuronal manipulation.

We first characterized the response of intact larvae to a mechanical stimulus: air-puff. We used a supervised learning algorithm that relies on a limited set of features associated to the shape, velocity and dynamical evolution of associated variable. The method was trained iteratively on a small number of larvae annotated manually, and is able to handle low resolution images of larva to specifically detect five action categories that we observed in response to air-puff: Hunch, Stop, Bend, Back-up and Forward Crawl. It can also detect a sixth action (Rolling), that doesn't typically occur in response to air-puff and is not a subject of this study. The actions are mutually exclusive and non-overlapping (S1 Fig). In order to ensure that this representation was stable in different genotypes and individual differences across the screen we applied the clustering to selected tested lines (S1 Fig). The stability of the representation throughout the screen (as well as in the lines of interest as seen in S1 Fig) suggest that representation of behavior as discrete action is not dependent on experimental conditions or on subset

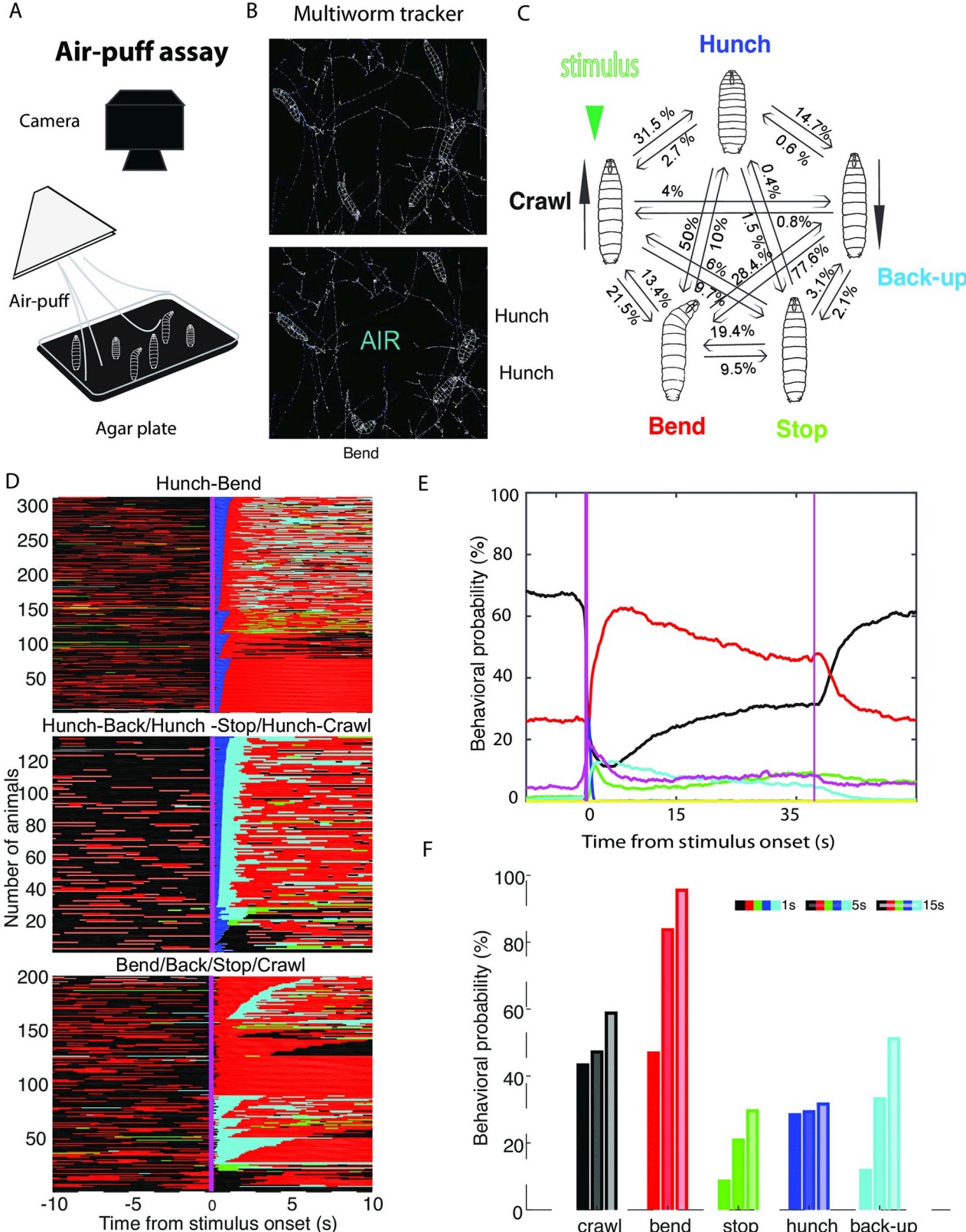

**Fig 1. Characterization of the behavioral response of wild-type larvae to an air-puff. A.** Behavioral set-up **B**. Still from a movie of contours of larvae acquired with the Multiworm tracker software **C**. In response to air-puff larvae perform a probabilistic sequence of five actions **D**. Ethogram of

the behavioral response to an air-puff (10s after stimulus onset). Different sorting of animals is shown based on the type of initial response. A subset of animals tested is shown for clarity for the different types of immediate responses and sequences. Each line is a larva. Different colors represent different actions: Blue-Hunch, Red-Bend B, Cyan-Back-up, Black-Crawl and in Green–Stop. *Top panel*: Hunch-Bend sequence, *Middle panel*: Hunch-Back, Hunch-Stop and Hunch-Crawl sequence, *Bottom panel*: Bend, Back, Stop or Crawl type of initial responses **E**. Behavioral probability (mean) before, during and after stimulation. Purple line depicts stimulus boundaries **F**. Behavioral probabilities in the first second, first 5 seconds and first 15 seconds after stimulus onset.

of larvae sharing common (unknown-unseen) properties. This natural clustering allows then associating to them various relevant amplitudes (velocity, angles of various part of the body, body length, derivatives of body length etc.) characterizing the action through continuous variables. Actions are also characterized by duration as the time spent within on cluster and naturally transitions from previous action to the current one and transitions from the current one onto the next. This description of behavior allows separating the nature of actions from all properties associated to them and thus allows probing selection processes during competitive interactions. Furthermore, within the current experimental paradigm, this description allows capturing the sequence of action generation where each discrete action has the same definition for all larvae. We equality minimized the effects of other factors on detection accuracy. For example, the tracking quality may affect the accuracy of detection (i.e. if the larva is tracked for very short periods of time and therefore tracks shorter than at least 20s were excluded from the analysis (see Material and methods)). Some actions may be ambiguous when they have very small amplitudes and durations and are therefore according to the definitions of actions used in this study fall in the small action category and therefore do not bias any of the categories. More details about the behavioral detection method can be find in the material and methods section of the paper including the unsupervised scoring of the approach allowing us to quickly evaluate the quality of the behavioral dictionary (S1C Fig).

We have previously studied and analyzed the immediate response of larvae to the air-puff using a length variation method [10,31] (see Material and methods for the link between new action definitions and previous ones). Here we analyzed a dataset of wild type larvae (control: w;;attP2-TNT larvae: progeny of male w;;attP2 crossed to female UAS-TNT-e flies (13 831 larvae), using the newly developed method and quantified previously undetected actions: Stops and Back-ups, in addition to Bends, Hunches and Forward crawls (Fig 1C–1E and 1G). Each of these actions occurs at different probabilities and these probabilities vary over the time of the response (Fig 1D–1F). For example, Hunches have a higher likelihood of occurring early in the response while the Stop occurs with a high probability early, its probability than drops and increase in the second half of the response (Fig 1E).

We computed behavioral probabilities for each of the five actions during first second, during five, and during fifteen seconds after stimulation (Fig 1F) and because the Hunches occur primarily immediately after stimulus onset, we focused the behavioral probability analysis of the screen data on the shortest time window. We further investigated the effect on air-puff intensity on the behavior. The probabilities of some actions vary significantly with stimulus intensity (i.e. higher Hunch and Bend and lower Stop at high intensity of air-puff) (S2A Fig).

In addition these actions occur in a probabilistic way. Different larvae will perform different actions in response to the same stimulus and repeated presentations of the stimulus give rise to different responses in the same larva (S3A–S3F Fig).

As actions occur in sequences one after the other, we further described the action sequences after stimulus onset. We computed transition probabilities between the different actions during the first three seconds after stimulus onset (Fig 1C, S11 Table) and found that from the Crawl the larvae will be transitioning strongly into a Bend or a Hunch (21.5% and 31.5%) respectively, less strongly into a Stop (9%) and with a lowest probability into a Back-up (4%).

Hunch can transition into each of the four other actions with the highest likelihood of transitioning into a Bend (50%) and lower probability of transitioning into the Crawl 2.7% and Stop (1.5%). The Bend has the highest probability of transitioning into a Back-up (28.4%). The Stop has a high likelihood of transitioning into a Bend (19.2%). From Back-up, the highest probability is to transition into a Bend (77.6%) and with lowest probability into a Hunch (0.6%). This suggests that the first action in a sequence is most frequently a Hunch or a Bend [10] and that if a Hunch is the first action it will be followed by the Bend with the highest probability. If Bend is the first action in the sequence than there is higher likelihood that the second action will be a Back-up (Fig 1C and 1D). Because transition probabilities between Bend and Back up are high in both directions, it suggests that there are multiple transition events between Bend and Back-up in one sequence with higher probabilities of transitioning from the Back-up onto the Bend. The transitions between Hunches and Bend, Hunches and Back-ups and Back-ups and Bends are asymmetrical, meaning that the transitions are more likely in one direction that the other (i.e. Hunches to Bend or Back-up than the other way around).

In summary, in response to an air-puff, the larvae can perform multiple types of actions. These can be organized in a probabilistic sequence where the Hunch is most likely to occur immediately after stimulus onset while the Bend, Back-up and Stop can occur later in the response. The behavioral response to an air-puff is thus well suited to use as an assay to study competitive interactions and transitions between multiple actions.

## Inactivation screen for mapping neurons and brain regions that underlie the behavioral response to air-puff

In order to identify best candidate neurons involved in competitive interactions and sequence transitions during mechanosensory responses, we applied the automated algorithm for behavioral detection to a behavioral inactivation screen dataset of larval responses to an air-puff stimulus. We analyzed behavioral responses of larvae in which we silenced small subsets of neurons and individual neurons. We tested 567 GAL4 lines that we selected for sparseness and expression quality based on an anatomical pre-screen out of the collection of the JRC GAL4 lines [20]. Expression patterns of the GAL4 lines used in this study are characterized in details and neuroanatomical information are available on http://www.janelia.org/gal4-gen1.

We determined families of hits by comparing the probabilities of performing each of the actions that occur in response to air-puff between test lines and the control. We considered as candidate for strong hits the lines that had significantly lower or higher behavioral probabilities in Hunches, Bending or Backing-up compared to the control (see Material and methods for details).

We then examined the different types of phenotypes (Fig 2B–2D, S1 Table) and found group of hits where: 1) one or more actions decreased, 2) one or more actions increased compared to the control and 3) one action was increased while (at least) another was decreased compared to the control. Based on these we determined three phenotype categories: 1) less actions, 2) more actions and 3) less of one action (or several actions) and more of another (or several other actions). We considered the latter category of hits as best candidates for implementing competitive interactions between mutually exclusive actions during sensorimotor decisions. These different categories of hits will be further discussed in sections: **Identification of central neurons involved in the air-puff response** and **Identification of central neurons and brain regions involved in competitive interactions.**

In addition to the behavioral probabilities, other features of behavioral responses, for example the amplitude of individual actions could also be affected. We quantified the amplitudes of Hunches and Bends (S2B and S2C Fig, S3 Table and S4 Table). The amplitude of Bending was

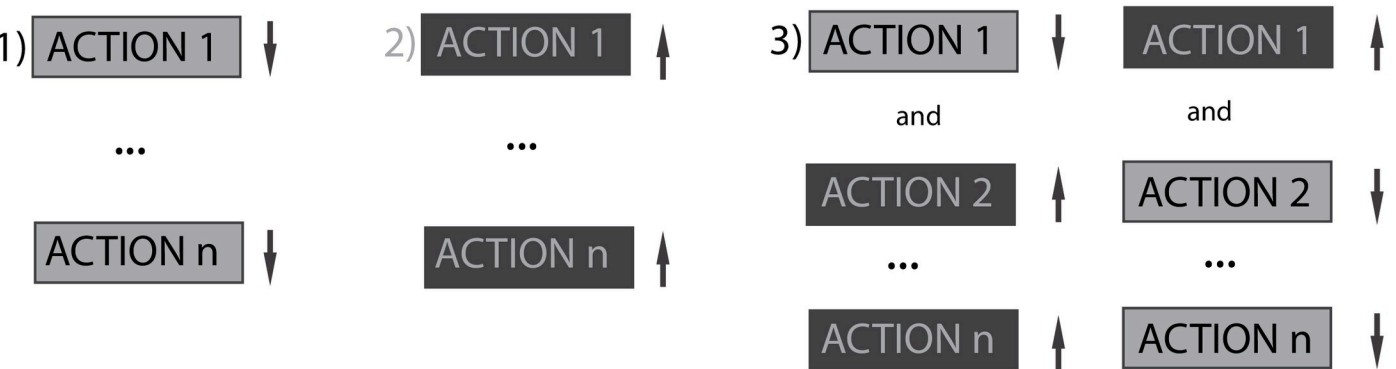

**A** Screen design

| 7000 lines JRC colection |

↓

| Anatomical pre-screen<br>Select lines with sparse expression patterns<br>in the larval brain and ventral nerve cord |

↓

| Screen 567 lines |

♂ X ♀
**567 GAL4 drivers**  UAS-TNT-e

↓

↓

| Detect changes in behavioral probabilities |

**B**

Bend probability (%) vs Hunch probability (%)

**C**

Back-up probability (%) vs Bend probability (%)

**D**

Hunch probability (%) vs Back-up probability (%)

High bend low another action probability,
High hunch low another probability
High back-up low another action  probability

High/low bend  probability
High/low hunch probability
High/low back-up probability

**E**  Behavior probability phenotype categories

1) ACTION 1 ↓
...
ACTION n ↓

2) ACTION 1 ↑
...
ACTION n ↑

3) ACTION 1 ↓   ACTION 1 ↑
and              and
ACTION 2 ↑       ACTION 2 ↓
...              ...
ACTION n ↑       ACTION n ↓

**Fig 2. Screen design and phenotype detection. A**. Experimental strategy and screen design **B-D.** Each dot represents average behavioral probability value for all the larvae of the same genotype. The red dot represents the w;;attp2>TNT control **B**. Scatterplot of Hunch probability against Bend probability for all the screened *Drosophila* lines. 10% of lines with lowest and highest probabilities of Hunch are shown in magenta. The lines with low Hunch and high Bend probabilities are shown in red. The lines with

high Hunch and low Bend probabilities are shown in blue **C**. Scatterplot of Bend probabilities against Back-up probability for all the screened lines. The 10% of lines with lowest and highest probabilities of Bends are shown in grey. The lines with low Bend and high Back-up probabilities are shown in cyan. The lines with high Bend and low Back-up probability are shown in red **D**. Scatterplot of Back-up probabilities against Hunch probability for all the screened lines. The 10% of lines with lowest and highest probabilities of Back-up are shown in light blue. The lines with low back-up and high hunch probabilities are shown in dark blue. The lines with high Back-up and low Hunch probabilities are shown in cyan. **E**. Behavioral probability phenotype categories: 1) less response phenotype: lower probabilities in one or more actions 2) more response phenotype: higher probabilities in one or more actions 3) competitive interaction phenotype: lower probability of one action and higher probability of at least one action or vice versa.

on average stronger in lines with a low probability of Bending. This could be due to the fact that it takes the animal longer to recover from the action and come back to neutral position. There were some exceptions. For example, the R15F02 that has both higher amplitude and probability of Bending. On the other hand, most of lines that hunch little also do weaker Hunches and those that do more Hunches do that more strongly. An exception to this is for example the R78A01 that has a high Hunch probability, but relatively low amplitude. Further investigating these outliers will help understand how the neurons control both the strength and probability of the response.

Because larvae usually crawl forward prior to the stimulation (it is a baseline foraging behavior) we excluded the Crawl from behavioral probability hit detection. Similarly, we didn't include the Stopping probability in the hit detection as the Stopping phenotypes cannot be interpreted unambiguously. At lower intensity of stimulation larvae stop more (S2A Fig) and therefore an increase in stopping (and decrease in one or more other actions) might reflect impaired sensing of air-puff. Indeed, this type of phenotypes can be observed when silencing chordotonals-key sensory neurons for sensing air-puff (S5 Table, see section: **At least two different types of sensory neurons mediate the air-puff response**). For all the hits, both the stopping and crawling probabilities are shown in S1 Table along with Hunch, Bend and Back-up probabilities used for hit-detection. These behaviors remain interesting to investigate and the data and probabilities presented in the supplemental material may provide the basis for studying in the future how and when these behaviors occur upon air-puff presentation.

Finally, some of the tested lines were significantly different to control prior to stimulus onset. This is most likely due to locomotor defects (S2 Table). We report these lines as these neurons are good candidates for examining circuitry underlying locomotion and posture. For the purpose of studying stimulus-evoked decisions, we excluded those lines from the analysis as their defects would interfere with animal's capacity to respond to stimulation.

## At least two different types of sensory neurons mediate the air-puff response

We have previously identified chordotonal sensory neurons as sensory neurons that mediate the air-puff induced responses as silencing of these neurons resulted in severely impaired responses [10,31]. We have applied the machine learning based behavioral detection method to analyze behavioral responses of larvae in which we silenced chordotonal sensory neurons using two drivers that we used previously R61D08 and iav and find lower probabilities of Hunching and Bending (Fig 3A, 3B and 3E) as expected. In addition, a lower probability of Backing up can be observed, that is significantly lower in iav>TNT larvae. While the response is severely impaired it is not completely abolished suggesting that some other sensory neurons could be involved in sensing air-current in addition to the chordotonals. In the *Drosophila* larva, there are multiple types of somatosensory neurons. Somatosensory neurons of type I, which are the monodendritic chordotonal and the external sensory neurons and multidendritic somatosensory neurons which can be subdivided into 4 classes: I, II, III, IV depending on the level of their arborizations and the position on the body wall [32–34].

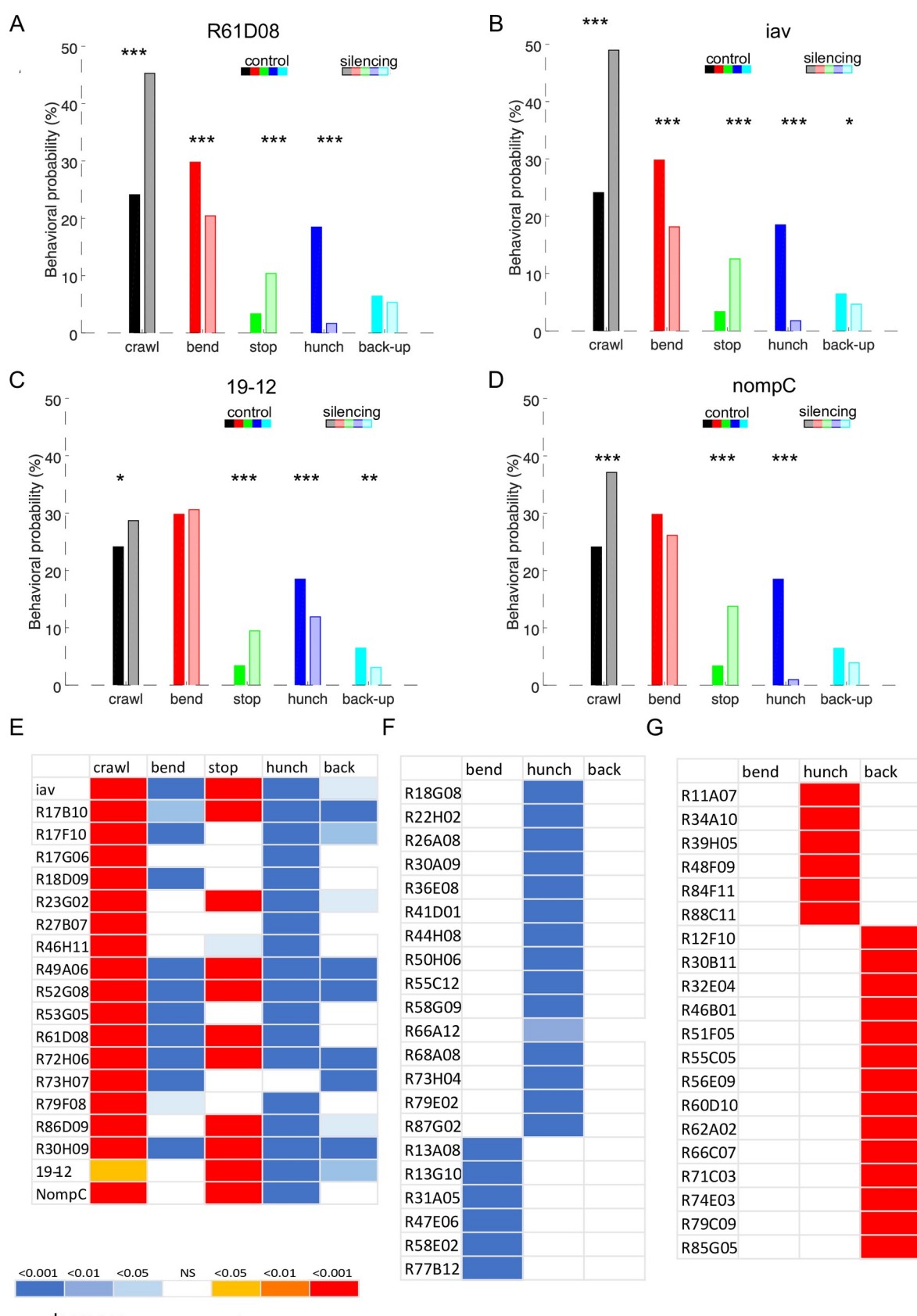

**Fig 3. Sensory and central neuron lines with affected behavioral probabilities. A-D**. Behavioral probabilities for sensory lines driving UAS-TNT-e. **A-B**. Using our behavioral detection method we find that silencing chordotonals with tetanus-toxin using iav and R61D08 drivers results in significantly less Hunches and less Bends in response to air-puff compared to the control, as we have shown previously [10,21,38] as well as less Back-up. **C**. silencing of md class III sensory neurons results in less Hunches and less Back-ups **D**. Silencing of md III and subsets of chordotonals results in less Hunches **E-G** Phenotypic summaries for the hits from the behavioral screen. The colors in the heatmap represent the p-values for each behavior for the comparison between each line shown to the right of each plot to the w; attP2-TNT control **E**. Sensory lines with phenotypes (19 lines). Known types of sensory neurons are shown in S5 Table **F**. CNS lines (41 lines). Hits with lower probabilities in one behavior compared to the control (21 lines). **G**: Hits with higher probabilities in one behavior compared to the control (20 lines). The behavioral probabilities and p-values can be found in S5 and S6 Tables and the S2 Data file.

Out of the lines screened we have identified 19 candidate hit lines that drive either only in sensory neurons (one or multiple types) or in sensory and some other neuron types (Fig 3E, S5 Table). We found that chordotonal sensory neurons are present in at least twelve GAL4 lines (S5 Table) including the chordotonal lines, R61D08 and iav-GAL4 lines that were described previously [10,31]. Silencing of chordotonal neurons using different drivers resulted in lower probability of Hunches and Bending and in some lines also of Backing-up. Differences between the lines could be due to the different strength of the Gal4 drivers and to the different subtypes of chordotonal neuron targeted by the drivers [34]. Multidendritic (md) neurons appear in some of the lines with perturbed response to the air-puff (i.e. R30H09, 19–12) and thus could be involved in air-puff sensing as well. We obtained a line that labels multidendritic class III (md III) neurons selectively. Md III have been shown to respond to light touch that induces similar response types: Hunches, Back-up, Bend, and Turns (away from the stimulus) [35,36]. We found that silencing of multidendritic neurons resulted in significantly less Hunches (although less pronounced phenotype than for chordotonals) and Backing-up, while the Bending response remained comparable to that in the control (Fig 3C, S5 Table). Hunching appears more severely impaired when silencing chordotonals then when silencing MD III sensory neurons (Fig 3A–3C and 3E) suggesting that air-puff induced Hunches are primarily mediated by the chordotonal sensory neurons. When silencing chordotonal and multidendritic class III neurons expressing tetanus-toxin using NompC (a TRP channel that confers light touch sensitivity [36,37]), a GAL4 driver that drives in both md III neurons and subsets of chordotonals, the larvae showed an impaired response with significantly less Hunches, and slightly, although not significantly, less Bends and Back-up, a phenotype consistent with a strong impairment in sensing air-current. Interestingly our recent study on anemotaxis revealed that multidendritic class III neurons may act synergistically with chordotonal sensory neurons to mediate larval navigation a wind speed gradient [38].

In summary, silencing of air-current sensing neurons resulted in lower probabilities of Hunches and for some lines also in Bending and/or Backing–up. When silencing the chordotonal neurons, the probability of Stopping was increased in several GAL4 lines (Fig 3, S5 Table), which suggested that the stopping response could be mediated by different sensory neurons and in absence of chordotonal sensory neurons this type of response becomes the primary response type. Alternatively, based on the finding that lower intensities of air-puff trigger more stopping than higher intensities, in absence of chordotonals, the larvae could stop more because they sense the stimulus less.

## Identification of central neurons involved in the air-puff response

As stated above, out of the lines driving in interneurons in the ventral nerve cord (VNC) and the brain, we observed three different categories of hits that have significantly different behavioral probabilities from the control (Fig 2B–2D). In the first two categories silencing of neurons results in significantly lower or higher probabilities of one or more actions compared to the control (Fig 2E, Fig 3B and 3C, Fig 4, S4 Fig, S6 Table, S7 Table), while in the third

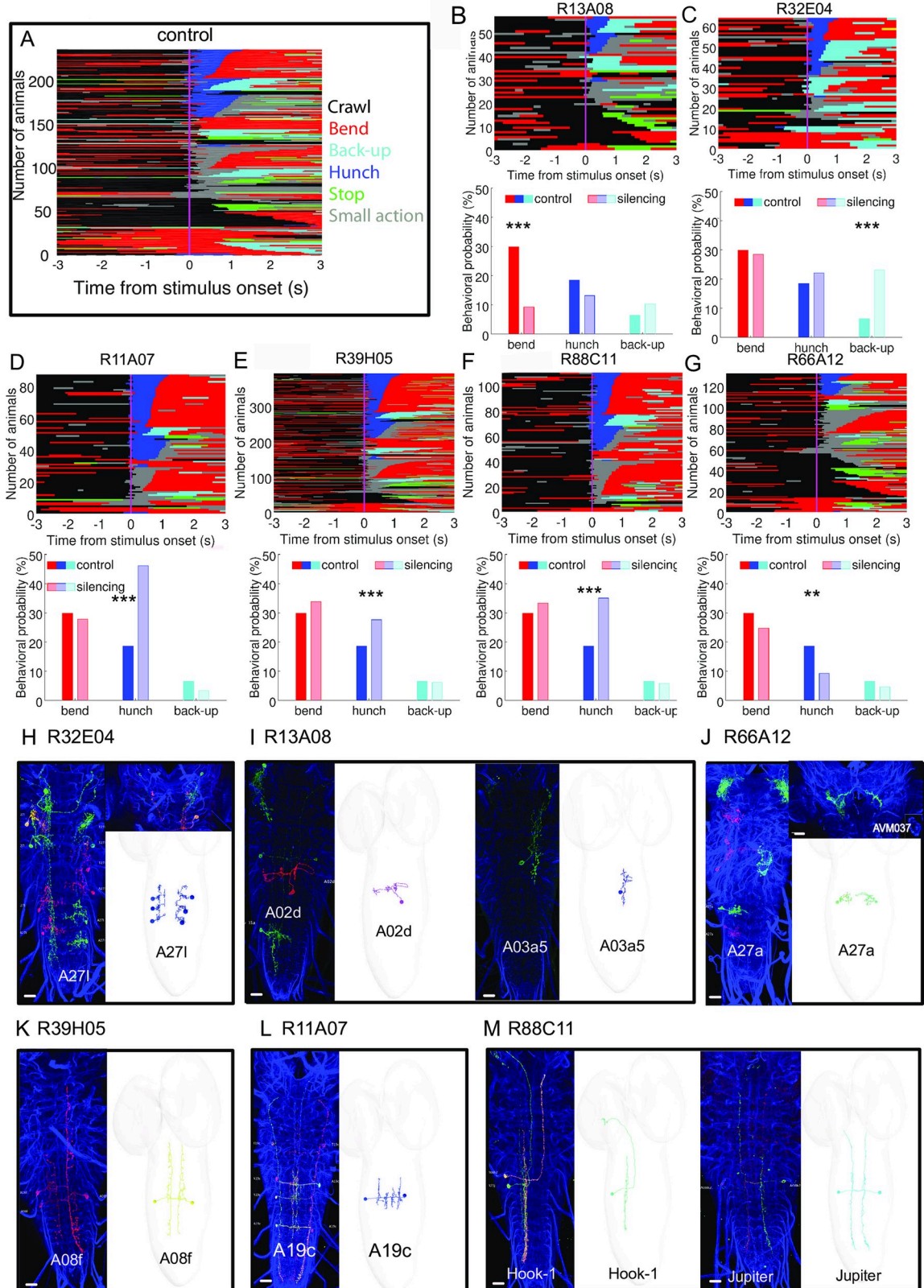

**Fig 4. Candidate first order interneurons and premotor neurons involved in mechanosensory responses. A** Ethogram of the air-puff response of randomly selected ca 200 control (w;; attP2-TNT) larvae for 3s prior and after stimulus onset. The larvae are sorted as a

function of the first and second action after stimulus onset. **B-G** Ethograms and behavior probabilities of different hit lines in which we silenced neurons using tetanus-toxin. For all of the lines ethograms for five actions Hunch-blue, Bend-red, Stop-Green, Back-up-Cyan and Crawl-Black are shown. The behavioral probabilities are shown for the first second after stimulus onset for Hunch, Bend and Back-up **H-M** Light microscopy images of larval brains with stochastic single-cell expression for the selected GAL4 in L3 (*left panels*) and the EM correlates (right panels) of the VNC neuron (reconstructed in a ssTEM L1 volume) are shown for the hit lines B-G. Scale bars are 20 μm.

silencing of neurons results in significantly higher probability of one action and a significantly lower probability of at least one action. We consider hits in this last category to be the top candidates for implementing competitive interactions between the different behaviors (Fig 2E, Fig 5, Fig 6, Fig 7, S5 Fig). These will be discussed in the next result section in more detail.

Out of the hit candidate lines from the primary analysis, we identified the strong candidates in the first two categories that are significantly different than the control. We identified 41 such hits where the probability of one action was significantly decreased or increased (Fig 3F and 3G, respectively, S6 Table).

Some of these lines have very sparse expression patterns that we analyzed in more detail. We used multicolor flip-outs to morphologically characterize the neurons in the lines whose silencing resulted in more Hunches (i.e. R11A07, R88C11, R39H05) (Fig 4D–4F, S4 Fig). We further identified or reconstructed them in an electron microscopy (EM) volume of the larval nervous system to reveal their connectivity (Fig 4K–4M, Fig 8, S6 Fig, S7 Fig, S1 Data file).

In two previous studies we have identified both local interneurons and ascending projection neurons involved in mechanosensory behaviors that receive inputs from chordotonal or multidendritic sensory neurons [10,38]. In a circuit for the choice between Hunching and Bending, we described distinct types of inhibitory neurons: LnA type interneurons that inhibit Basin-2 neurons (that promotes Bending and inhibits Hunching) and disinhibit Basins-1 projection neuron (that promotes Hunching and Bending), while LnB interneurons inhibit Basin-1 and disinhibit Basin-2. FbLN receive input from Basin neurons and inhibit the Ln neurons thus disinhibiting the Basins [10] (Fig 8, S1 Fig).

By matching light microscopy to electron microscopy images, we found that R39H05 drives in the A08f neuron that we previously shown to synapses onto the Ladder-e (LnB neuron type) in the behavioral choice circuit for Hunch and Bend [10]. Upon analysis of the connectivity (S1 Data file) we found that in addition to synapsing onto the local inhibitory neurons in the behavioral choice circuit it also synapses onto the A10j neuron that receives input from Basins [21] (see section **Identification of central neurons and brain regions involved in competitive interactions**). In another line (R88C11), we found two neurons that receive input from chordotonals. One is Jupiter (A08h2) that we previously shown was required for anemotaxis [38]. The second one is a long-range projection neuron A23j, that we reconstructed in the EM volume and named Hook-1. This neuron receives input from chordotonals, Ladders (mostly Ladder-c type) [10] and Jupiter and projects to higher order regions of the nervous system. Finally, in a third line (R11A07) we identified a neuron A19c in the motor domain [21] that receives input from Basin-2,4. Thus, all of the three sparse lines drove in neurons that are related to chordotonal circuitry. This further validates the candidate neurons as being involved in mechanosensory responses.

Interestingly, in the line with a significantly higher Backing-up probability (R32E04) compared to the control (Fig 4C, S4 Fig) we identified the premotor neuron A27l [28] that could therefore be involved in inhibiting Backing-up (Fig 4H). The authors in [28] find that the inhibitory A27l neuron receives input from a MDN brain descending neuron that can induce Backward locomotion following optogenetic activation and speculate that this could be achieved through disinhibition (by inhibition of the inhibitory A27l neuron). The phenotype

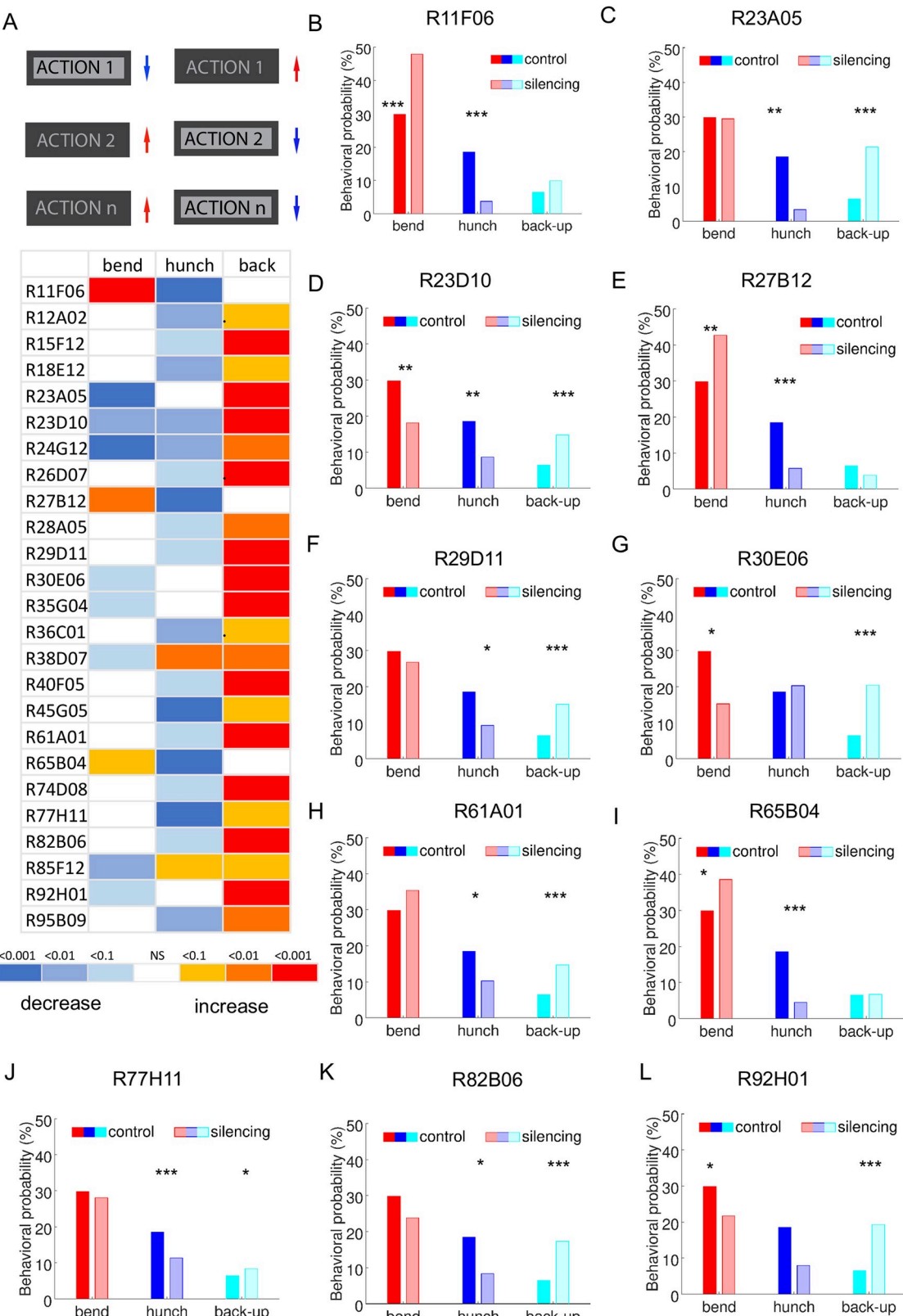

**Fig 5. Competitive interaction hit lines. A**. *Top panel*: schematic of the competitive interaction phenotype. *Bottom panel*: Heatmap. Phenotypic summaries using selected behaviors (Hunch, Bend and Back-up) for the "competitive interaction" hits from

the behavioral screen as depicted in the top panel. The colors in the heatmap represent the p-values for each behavior for the comparison between each line to the w;;attP2-TNT control as shown in the p-values legend. The hits that have lower probabilities in at least one action and higher probabilities in at least one other action (25 lines) are shown **B-L** Probabilities for Bending, Hunching and Backing-up for 11 lines selected based on their neuronal expression patterns (see Fig 6) compared to the control w;; attP2-TNT control. *:p<0.1, **:p<0.01,***:p<0.001.

that we find: increase in probability of Back-ups upon silencing of neurons is consistent with this hypothesis.

Silencing of neurons in R13A08 resulted in lower probabilities of Bending (Fig 4B, 4I, S4 Fig). By matching the light microscopy images to the EM images, we identified the A03a5 and A02d neurons. A03a5 [27,29,39] based on connectivity data is a pre-motor neuron as it synapses on multiple motor neurons, while the A02d (Fig 4I) is a glutamatergic neuron that receives input from proprioceptive neurons and synapses on premotor neurons [27]. It can be observed in the ethogram (Fig 4B) that the decrease in Bending probability in this line is due to both the decrease in Bending occurring as the first action upon stimulus onset as well as the second action upon transition from the Bend compared to the control. The Back-up probability is slightly although not significantly increased. We cannot exclude that the neurons in this line could thus also be involved in the Back-up. A sparse line (R66A12) drove in the A27a neuron [10,27] in the VNC and a brain neuron. A27a neuron receives input from a subtype of chordotonal sensory neurons, proprioceptive neurons and the A27l premotor neuron (and weakly from LnA, LnB and FbLN neurons). Investigation of its downstream connectivity revealed partners in the chordotonal circuitry: it strongly synapses onto FbLN, LnB neurons and on Basin-1 and weakly on LnA. Silencing of this neuron using the R66A12 driver resulted in significantly less Hunching (Fig 4G and 4J) and slightly but not significantly less Bending and Backing-up.

Interestingly in these sparse lines some of the neurons are downstream of mechanosensory chordotonal neurons (Jupiter, Hook-1, Baton, A27a), some on the motor side (A19c, A27l, A03a, A02d) consistent with the role of these neurons in sensorimotor behaviors.

In some of the lines we silenced multiple neurons and we cannot exclude that the phenotype results of inactivation of both neurons simultaneously. It will be interesting to test when possible individual neurons and see whether the phenotype is due to silencing one of the neurons or both (or three neurons). This for example could be the case for the R88C11 line as it labels Jupiter and Hook-1 neurons that both receive input from chordotonals.

## Identification of central neurons and brain regions involved in competitive interactions

We determined that the detected actions in response to air-puff are mutually exclusive (S1 Fig) and therefore we were especially interested in identifying best candidate neurons that could implement competitive interactions between these actions. We reasoned that if a neuron is required for selecting between mutually exclusive behaviors, inactivating that neuron might significantly increase the probability of one action, at the expense of another or several other actions.

We identified 25 hits in which larva perform significantly less of at least one action and significantly more of at least another action (Fig 5, S5 Fig, S10 Table). This phenotype could be observed with different combinations of actions: Hunches, Bends and Back-ups. We never observed a decrease in Back-up probabilities and increase in Bend and/or Hunch probability.

Some of these 25 lines have sparse neuronal expression patterns in one to four neuron types. We morphologically characterized the neurons in these lines using multicolor flip-out

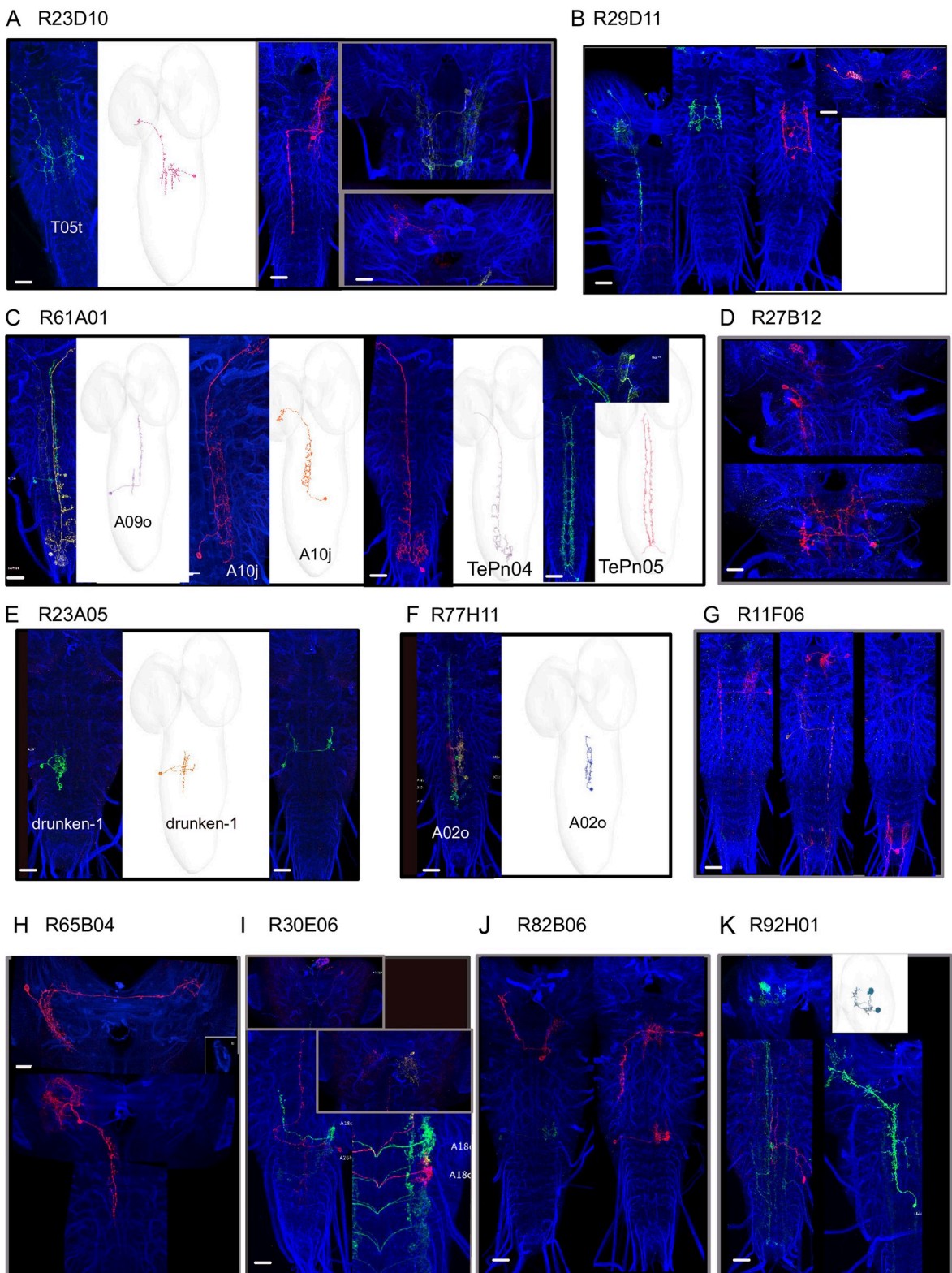

**Fig 6. Candidate neurons for competitive interactions as revealed by multicolor flip-out (MCFO) and EM reconstruction images.**
**A-K** Light microscopy images of larval brains with stochastic single-cell expression for the selected GAL4 in L3 (*left panels*) and the EM

correlates reconstructed in a ssTEM L1 volume for the identified neurons in the lines shown in A, C, E-F, K. EM that have been matched with MCFO images from the hit GAL4 lines. B, D, G-K. The R11F06 driver labels a neuron in the terminal segment, a thoracic neuron with a contralateral projection and a neuron in the SEZ (suboesophageal zone). The R27B12 line labels two interneurons in the SEZ, while the R82B06 labels two neurons in the SEZ one projecting to the brain, the other descending in the VNC. The R65B04 labels two brain neurons: one with a contralateral projection, one descending in the VNC. The R24G12 line labels an abdominal interneuron and an SEZ interneuron. Finally, the R92H01 line labels a thoracic ascending neuron T12u and in addition to the MBON- c1.Scale bars are 20 μm.

[40] and determined their connectivity patterns by identifying them in the electron micros-copy volume that spans the larval nervous system [21] (Fig 6A–6K).

In lines: R23D10, R77H11, R23A05, R29D11, R61A01 we were able to match some of the neurons from the multicolor flip-out images to the EM reconstruction images and analyze their connectivity to each other and to previously identified elements of the mechanosensory network [10,21,38] as well as nociceptive and pre-motor networks [21,30,39,41]. This is sum-marized and shown in Fig 8. and S6 Fig.

In the R23D10 we were able to identify the T05t neuron in the EM volume [21] and found it receives input from LnA type interneurons and Basin neurons in the Hunch/Bend behav-ioral choice circuit [10] (Fig 8). In the R61A01 line we identified four different neurons: A10j, A09o, TePn04 and TePno5. The A10j neuron [21] receives input from LnA (Drunken-1, Grid-dle-1,2) LnB (Ladder-a) and Basins neurons and synapses strongly on the T05t neurons and the FbLN (feedback interneurons) that are involved in sequence transition and maintenance [10]. The A09o neuron [41] receives inputs from multidendritic class III and Wave neurons [21,39] as well as projections neurons along the nerve cord. The TePn04 neuron [21], receives input from Basin-2 and LnA and LnB neurons and ascends to the higher order regions while the Tepn05 neuron receives input from multidendritic class III (and IV) sensory neurons and related interneurons (i.e A09e, DnB, Wave [30,38,39], as well as FbLN neurons and synapses onto Basin-2 and 4, the Wave and A09o neurons and projects to higher order regions [30,41]. A sparse line (R77H11, Fig 6F) drove in a A02o neuron also called Wave based on the shape of its axon [21,39]. This neuron receives inputs from multidentritic class III and IV neurons, Basins, synapses onto A09o and was shown to mediate touch induced responses. Its down-stream connectivity revealed partners in the motor domain, including the A03a5 premotor neuron [39] that we find was in the R13A08 line with a reduced bending phenotype. Silencing of this neuron using the R77H11 driver resulted in significantly less Hunching and more Back-ing-up compared to the control (Fig 5J).

Additional lines drove expression in neurons that received input from chordotonals (R29D11, a thoracic ladder neuron, R23A05-drunken-1 [10]). Some of these lines also drove expression in brain neurons, like for example the R92H01- that drove expression in MBON-c1, a mushroom body output neuron [25].

We were not able to find some of the neurons from the sparse lines in the EM volume. Nev-ertheless, the multicolor flip-out analysis allowed to analysis the expression patterns and visu-alize single neuron types in the ventral nerve cord and brain in these lines (R11F06, R27B12, R30E06, R65B04, R82B06, Fig 6D, 6G–6K).

The multicolor flip-out images with the example of all the neurons we were able to obtain are shown in Fig 6 along with EM images in the case we were able to identify them in the EM volume. However, if a neuron was stochastic or very weakly express we exclude it from the image as these neurons are likely not causing the phenotype. The complete list of the neuronal annotations for all the candidate hit lines can be found in S8 Table.

Altogether, these data show the candidate neurons for competitive interactions are located in different areas of the nervous system (Fig 6A–6K, Fig 8), suggesting that the different sites of competitions are distributed in different regions of the nervous system: abdomen, thorax

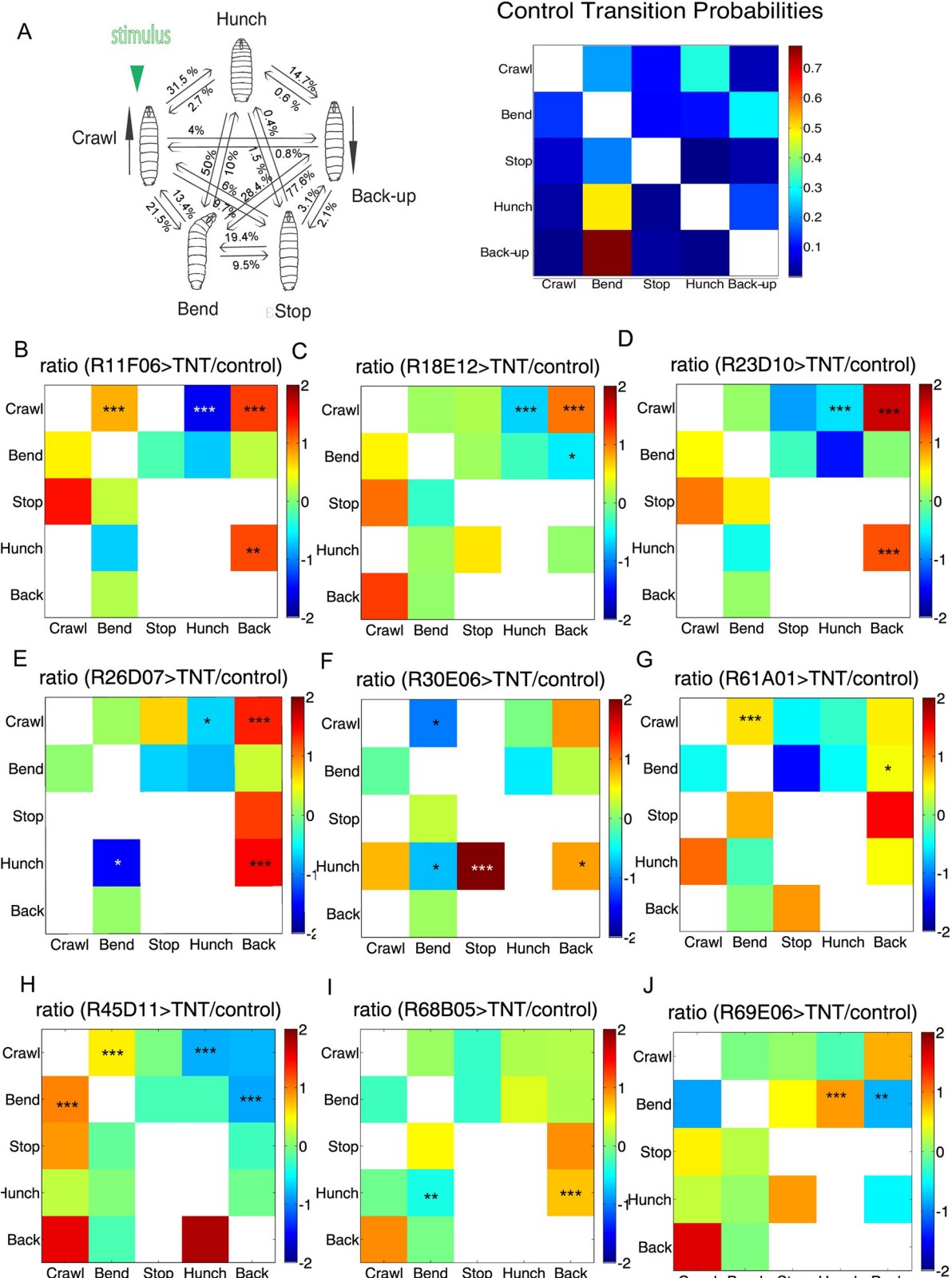

**Fig 7. Transition probabilities phenotypes. A**. Transition probabilities for the w;; attP2-TNT control **B-J.** ratio of transition probabilities for selected hit lines relative to the control **B-G** Examples of transition probability phenotypes for hit lines of competitive interaction hits (Fig

5). In addition to transitions from the Crawl to the initial action after stimulus onset (Hunch, Bend and/or Back-up) these lines also have transitions probabilities between later element in the sequence affected. R11F06 and R23D10 (A, C) have higher transition probabilities from Hunch to Back-up (additional lines with this phenotype R29D11, R36C01, R38D07, R77H11 are shown in S11 Table). R26D07 (E) had decreased transition probabilities from Hunch to Bend, R30E06 (F) shows decreased Hunch to Bend and increased Hunch to Back transition probabilities. For the R18E12 (C) and the R61A01 (G) lines, the Bend to Back transition probabilities were decreased and increased respectively. **H-J**. Example of transition probability phenotypes from lines screenwide *: p<0.05, **: p<0.01, ***: p<0.001 (chi-square test).

and brain. The pathways from the chordotonal and multidendritic class III neurons go to the pre-motor regions or ascend in multiple parallel pathways to higher regions of the nervous system.

## Neurons and brain regions underlying sequence transitions

We have shown that the individual actions Hunch, Bend, Stop, Back-up, and Crawl can be organized in probabilistic sequences of actions (Fig 1, Fig 7A). Competitive interactions could occur during selection of the initial response to the stimulus or between the different actions in the sequence to establish the order of the individual elements.

In order to determine whether the competitive interaction candidate hit line phenotype are consistent with the role of its neurons in selecting the initial response to the stimulus or selecting the next action in the sequence (or both), we computed transition probabilities between each pair of individual actions in the candidate hit lines (S11 Table). If a neuron implements the selection of the initial response, its silencing would result in increased transitions from the baseline crawling behavior to Hunch, Bend and/or Back-up. On the other hand, if a neuron controls the selection between the later elements in a sequence its silencing would result in perturbed transitions between the different individual actions Hunch, Bend and Back-up. As expected, in the 25 candidate competitive interaction hit lines with affected behavioral probabilities in first second of stimulation (Fig 5, Fig 6A–6K) the transition probabilities from the Crawl (the dominant baseline behavioral state previous to stimulus onset) to Hunch, Bend and/or Back-up were also perturbed (Fig 7, S11 Table) and these two statistics are intricately linked. This confirms that the neurons in these lines are candidate neurons for competitive interactions during selection of the initial response to the stimulus. In a subset of these 25 hit lines the transitions between Hunch, Bend, and Back-up were also affected, suggesting they also play a role in selecting the later elements of the sequence (Fig 7, S11 Table).

Sequence transition could also be affected in lines that were not among the best 25 hit lines for competitive interactions between Hunch/Bend/Back-up at the start of the sequence (just upon stimulus onset) but later during the response. Such lines would not come up in our competitive interaction hit lines that were chosen based on their phenotype in the first second upon stimulus onset. We therefore analyzed transition probabilities between all the five actions: Crawl, Bend, Stop, Hunch and Back-up in all the lines in the screen to look for additional candidates that would be involved in controlling transitions between behaviors in the air-puff induced sequence (Fig 7B–7I, S11 Table). Among the screened lines we identified five central brain neuronal lines with lower Bend to Back-up transition probabilities (S11 Table). For example, in the R45D11 line shown in Fig 7G, the transition from Bend to Back-up are decreased, while Bend to Forward Crawl transitions are increased. This phenotype is consistent with a role of the candidate neurons in the proper sequence progression by promoting transitions from Bend to Back–up and inhibiting reversals from Bend to Crawl. We also identified twelve lines with an increase in Hunch to Back-up transition probabilities, while Hunch to Bend transition probabilities were decreased. These lines could be implementing asymmetric competitive interactions between Bend and Back-up to select the next action in the sequence after the Hunch. The R68B05 example is shown in Fig 7H. Finally, we found two

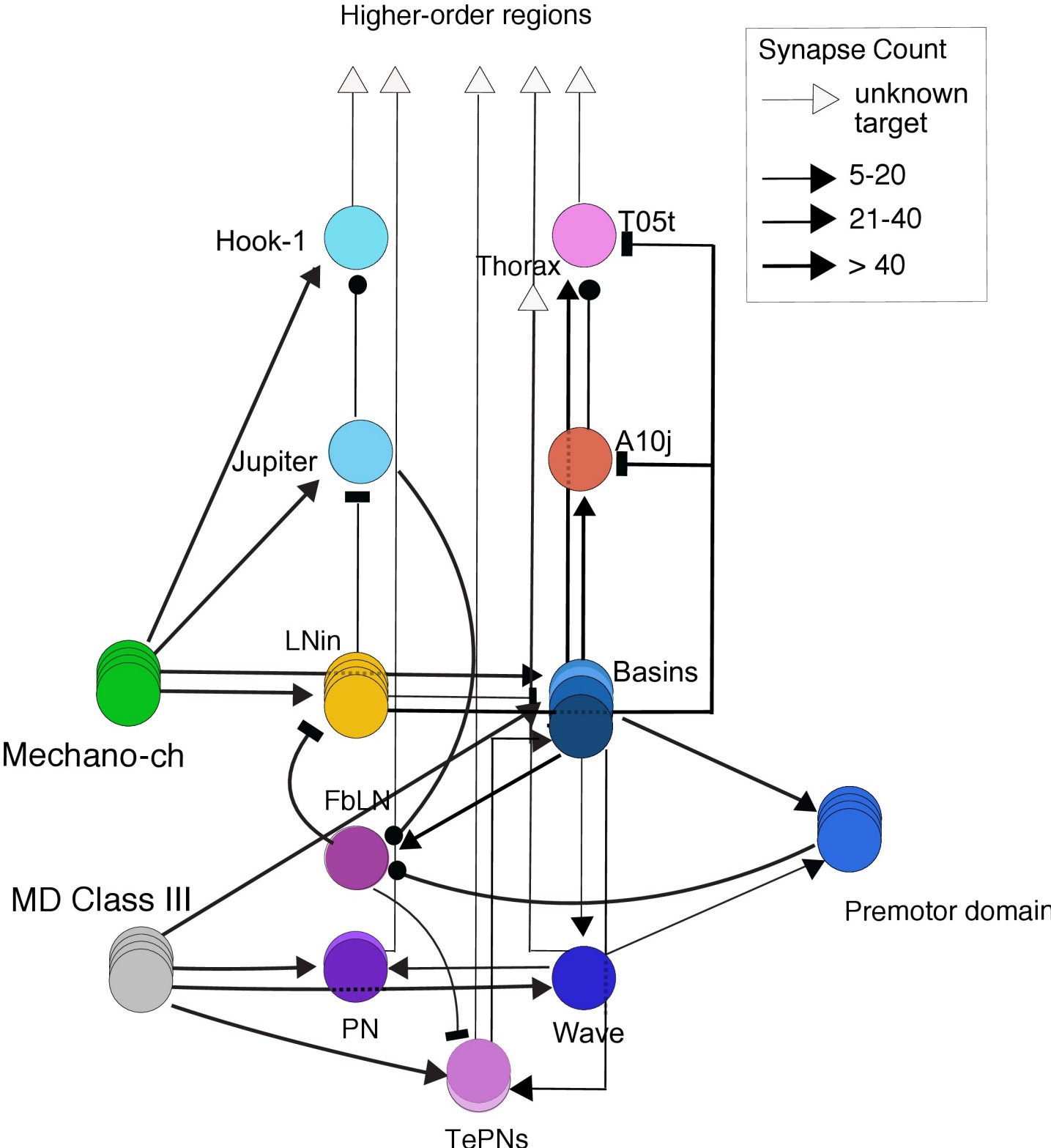

**Fig 8. Putative pathways in the mechanosensory network.** –a summary diagram of the main pathways between the neurons identified in the hit lines in this study and some previously published key neurons involved in mechanosensory responses: Basin-1-4 (B1-B4), LnA, LnB, fbLN, A09e [10,21,38]. Connectivity is based on previously published reconstruction and newly reconstructed connections (S1 Data file). The arrows indicate excitatory connections, T-bars indicate inhibitory connections and lines with black circles indicate when it is unknown whether the connection is excitatory or inhibitory. The width of the line is proportional to the strength of the connection. Some connections are not shown for clarity (all connections can be found in the connectivity matrix, S1 Data file).

lines with increased transition probabilities from Bend to Hunch, while Bend to Back-up transitions were decreased (R69E06, Fig 7I, S11 Table). The neurons in this line could thus be preventing reversals from Bend to Hunch and ensuring proper progression of the sequence from Bend towards the Back-up (next action in the sequence).

These phenotypes reveal that some of these neurons (i.e. in lines R45D11, R69E06) might be involved in maintaining a state (preventing reversals on previous state) and promoting transitions onto the next state. They would be thus ensuring the proper progression of the sequence, while others (i.e. in the line R68B05) might be involved in asymmetric competitive interactions between transitions from Hunch to Bend and Hunch to Back ensuring that Bend is more likely to occur after the Hunch then the Back-up (Fig 1C).

## Discussion

In order to identify neurons and brain regions underlying competitive interactions and transitions between actions during mechanosensory responses we performed a high-throughput inactivation screen where we silenced individual neurons and groups of neurons (using tetanus-toxin) in 567 genetic GAL4 lines in *Drosophila* larva and looked at the effects of these manipulations on larval behavioral responses to a mechanosensory cue.

We characterized the behavioral response of wild-type larvae to the stimulus (air-puff) and found that larvae perform a probabilistic sequence of five different actions. We developed and used an automated approach that detects and distinguishes five different discrete behaviors that larvae perform in response to the air-puff. Evidences suggest that the discrete action description is relevant when compared to a continuum approach as parameters associated to larva dynamics tend to naturally cluster (S1 Fig). The representation is found to be stable even for large number of larvae while their characteristics (amplitude of actions, duration, size of the larva, shape etc.) can vary significantly. Yet we point out that it does not mean that all behaviors and actions that larvae exhibit are necessarily described as only discrete actions.

We used this analysis to describe phenotypes that result from manipulation of different populations of neurons or single neuron types. We find phenotypes that are consistent with a specific role of neurons in sensory processing or motor control, competitive interactions, and sequence transitions. Neuronal expression data for all of the GAL4 lines used in this screen have been previously published [20]. The number of neurons that were targeted in our tested lines varies from 1 to 7 pairs on average and smaller number of the GAL4 lines the driver is restricted to a single neuron type. We analyzed the morphology of top hits in more detail using single-cell FLP-out and we analyzed their connectivity using electron-microscopy reconstruction (Fig 8).

We developed a framework for selectively identifying circuit elements underlying competitive interactions and sequence transitions. Sensory-processing, sensorimotor decisions, and sequence generation are intertwined processes as the latter two will depend on how the sensory information is processed, and the sequence production mechanistically might depend on competitive interaction between distinct actions as suggested by models of sequence generation like competitive queuing or chains of disinhibitory loops [10–12,42]. Nevertheless, we used the reasoning below to identify neurons selectively involved in competitive interactions that underlie sensorimotor decisions and sequence generation.

We reasoned that, if the stimulus cannot be processed and thus perceived accurately the animals might respond less, by performing less of all or some of the actions. If the sensory processing is affected in the opposite way (hypersensitivity), animals might respond more, and perform more of all or some of the actions normally observed. Thus, the neurons that gave such inactivation phenotypes (of less of one or more actions; or more of one or more actions)

could be involved in any aspect of sensory processing or motor control. We also cannot exclude that these larvae responded less because the inactivation of the neurons modulates the overall animal state.

However, inactivation of neurons involved in mediating competition between actions is expected to result in increased probability of one action and a decreased probability of one or more other actions (or the converse) as the neuron implementing the competitions will promote one action while inhibiting competing options. Based on this logic, we identified 25 hits (GAL4 lines) that were top candidates for selectively mediating affected competitive interactions. We characterized morphologically the neurons in these lines using light microscopy of multicolor flip-out and for some of the neurons determined their connectivity by identifying them in the electron microscopy volume. We found that some of these neurons received input from chordotonal sensory neurons, chordotonal related interneurons or multidendritic class III sensory neurons while others were pre-motor neurons. In addition, we found other neurons that project to or are located in the brain (Fig 8, S6 Fig, S7 Fig). Taken altogether, the GAL4 lines that we identified as hits drive in neurons that are located in the ventral nerve cord (both abdomen and thorax region), suboesophageal ganglion and brain (Fig 8, S6 Fig, S7 Fig). This suggests that the networks for competitive interactions between actions that occur in response to air-puff are distributed across the nervous system.

The idea that sensorimotor decisions are made "through a distributed consensus that emerges in competitive populations" and that interactive behaviors require sensorimotor and selection system to function in parallel [3,10,43] have emerged in various fields [1,2,5,44–46], but it has been challenging to elucidate the neuronal architecture that would implement such sensorimotor decisions. The *Drosophila* larva, because of its numerical simplicity, small size and the existence of multiple experimental approaches for structural and functional connectivity studies, behavioral genetics, optogenetics etc. is an ideal system for investigating how the outcomes of these competitive interactions at the different sites are integrated across the nervous system to give rise to coherent sensorimotor behaviors.

The neural architecture that controls the productions of probabilistic action sequences and establishes the order of the individual elements in the sequence is also poorly understood. Here we identified a number of hit line phenotypes that were consistent with an implication of the neurons in ensuring proper ordering of individual elements in the sequence. For example, the neurons in the R45D11 line could be inhibiting reversals from Bend to Crawl and promoting transitions from Bend to Back, while neurons in the R69E06 line could be promoting transitions from Bend to Back-up while preventing reversals from Bend to Hunch. In our previous work on a two- element Hunch-Bend sequence in response to an air-puff, we have proposed that transitions to the next element in the sequence and reversal to the previous element are controlled through two different motifs: lateral disinhibition from the neuron driving one behavior onto the neuron driving the following behavior and feedback disinhibition that provides a positive feedback that stabilizes the second behavior and prevents reversals back onto the previous actions [10]. We speculated that chains of such disinhibitory loops could be a general mechanism for generating longer behavioral sequences. In the case of longer sequences (more than two elements) the maintenance of a selected action (through a positive feedback) after the transition from the previous action has occurred would need to be balanced with promoting the transition from the current onto the following action in the sequence. The candidate neurons in the R45D11 and the R69E06 could represent a starting point for investigating these mechanisms as their phenotype are consistent with preventing reversals from Bend to Hunch and Crawl and promoting transitions from Bend to Back-up (that represent nearly 80% of transitions from Bend). Another category of phenotypes, increase in transitions from Hunch to Back-up and decreased from Hunch to Bend, suggests that asymmetric competitive

interactions exist between transitions to Bend and Back-up (from Hunches) where the transitions from Hunch to Bend inhibit transitions from Hunch to Back-up but not the other way around. Such a mechanism would allow a progression of a sequence in a probabilistic way where the transitions from Hunch to Bend are more likely (50%) than to Back-up (less than 15%).

In summary, our screen provides a roadmap for investigating the neural circuit mechanisms underlying the different computations during mechanosensory responses. It also offers a starting point for identifying the mechanisms underlying the competitive interactions between behaviors as well as the transition between individual actions in probabilistic sequences across the nervous system. While the number of neurons that were targeted in our tested lines varies from one to seven pairs on average, and sometimes more, in the case when the lines label multiple neuron types, intersectional strategies can be used to further refine the expression patterns. In the larva, a volume of electron microscope data has been acquired and more than 60% of the nervous system has been reconstructed through collaborative efforts [10,21,22,24,25,27,29,47–49]. The synaptic partners of the identified candidate neurons can therefore be further reconstructed in the electron microscopy volume. Combined with EM reconstruction, physiology, and modeling the candidate lists of neurons can be used to relate circuit structure and function across the nervous system and unravel the principles of how the nervous system selects actions and produces action sequences in response to external stimuli.

## Materials and methods

### Drosophila stocks

We screened 567 GAL4 lines. 564 lines were from the Rubin collection lines listed in S3 Data file (available from Bloomington stock center) each of which is associated with an image of the neuronal expression pattern shown at http://flweb.janelia.org/cgi-bin/flew.cgi. In addition we used the insertion site stocks, w;attP2 [50], OK107GAL4, 19-12-GAL4, NompC [51] and iav-GAL4 [52]. We used the progeny larvae from the insertion site stock, w;;attp2, crossed to the appropriate effector (UAS-TNT-e (II)) for characterizing the w;; attP2 were selected because they have the same genetic background as the GAL4 tested in the screen. We used the following effector stocks: UAS-TNT-e [53] and pJFRC12-10XUAS-IVSmyr::GFP (Bloomington stock number: 32197).

### Larva dissection and immunocytochemistry

To analyze the expression pattern of the GAL4 lines, we crossed the lines to pJFRC12-10XUAS-IVSmyr::GFP (Bloomington stock number: 32197) [20]. The 'FLP-out' approach for stochastic single-cell labelling will be described in detail elsewhere [40]). In brief, heat-shock induced expression of FLP recombinase was used to excise FRT-flanked interruption cassettes from UAS reporter constructs carrying HA, V5, and Flag epitope tags, and stained with epitope-tag specific antibodies. This labelled a subset of the cells in the expression pattern with a stochastic combination of the three labels. The progeny larvae were placed in a phosphate buffered saline (PBS; pH 7.4) and fixed with 4.0% paraformaldehyde for 1–2 hr at room temperature, and then rinsed several times in PBS with 1% Triton X-100 (PBS-TX). Tissues when then mounted on poly-L-lysine(Sigma-Aldrich) coated coverslips and then transferred to a coverslip staining JAR (Electron Microscopy Sciences) with blocking solution, 3% normal donkey serum in PBS-TX for 1 hr. After incubation, the tissue was rinsed for several hours in PBSTX, and dehydrated through a graded ethanol series, cleared in xylene and mounted in DPX (Sigma) Images were obtained with 40x oil immersion objective (NA 1.3) on a Zeiss 510 Confocal microscope. Images of each nervous system were assembled from a 2xarray of tiled stacks, with each stack

scanned as an 8-bit image with a resolution of 512x512 and a Z-step of 2 μm. Images were processed using Fiji (http://fiji.sc/).

## Electron microscopy reconstruction and wiring diagrams

EM reconstruction was performed using a complete CNS serial section transmission EM volume from a 6-hour old *Canton S G1 × w^1118 [5905]* larva, with a resolution of 3.8nm x 3.8nm x 50nm [21]. Reconstruction and synapse annotation followed previous protocols [22] using the web-based annotation software, CATMAID [54]. To identify neurons of interest covered by our GAL4 lines, we examined the reconstructed neurons in the CATMAID database. Gross morphologies, including axon bouton, dendrite, and cell body positions, were compared to light microscopy images to positively identify neurons in EM. We thus identified previously reconstructed neurons (see S9 Table for full list of neurons and corresponding references). To identify previously unpublished neurons, we performed exploratory tracing downstream of mechanosensory chordotonals. The identified neurons were fully reconstructed, including all synapses and fine dendritic processes. If hemilateral pairs of neurons received at least 3 synapses from a particular neuron or on both sides, we considered them strong neuronal partners.

## Behavioral apparatus

The apparatus was described previously [10,31]. Briefly, the apparatus comprises a video camera (DALSA Falcon 4M30 camera) for monitoring larvae, a ring light illuminator (Cree C503B-RCS-CW0Z0AA1 at 624 nm in the red), a computer (see [31] for details); available upon request are the bill of materials, schematic diagrams and PCB CAM files for the assembly of the apparatus) and a hardware modules for controlling air-puff, controlled through multi worm tracker (MWT) software (http://sourceforge.net/projects/mwt) [55], as described in [31]. Air-puff is delivered as described previously (Ohyama et al., 2013). Briefly it is applied to a 25625 cm2 arena at a pressure of 1.1 MPa through a 3D-printed flare nozzle placed above the arena (with a 16 cm 6 0.17 cm opening) connected through a tubing system to plant supplied compressed air (0.5 MPa converted to a maximum of 1.4 MPa using a Maxpro Technologies DLA 5–1 air amplifier, standard quality for medical air with dewpoint of 210uC at 90 psig; relative humidity at 25uC and 32uC, ca. 1.2% and 0.9%, respectively). The strength of the airflow is controlled through a regulator downstream from the air amplifier and turned on and off with a solenoid valve (Parker Skinner 71215SN2GN00). Air-flow rates at 9 different positions in the arena were measure with a hot-wire anemometer to ensure even coverage of the arena (Extech Model 407119A and Accusense model UAS1000 by DegreeC). The air-current relay is triggered through TTL pulses delivered by a Measurement Computing PCI-CTR05 5-channel, counter/timer board at the direction of the MWT. The onset and durations of the stimulus is also controlled through the MWT.

## Behavioral experiments

Embryos were collected for 8–16 hours at 25˚C with 65% humidity. Larvae were raised at 25˚C with normal cornmeal food. Foraging 3 instar larvae were used (larvae reared 72–84 hours or for 3 days at 25˚C).

Before experiments, larvae were separated from food using 10% sucrose, scooped with a paint brush into a sieve and washed with water (as described previously). This is because sucrose is denser than water, and larvae quickly float up in sucrose making scooping them out from food a lot faster and easier. This method is especially useful for experiments with large number of animals. We have controlled for the effect and have seen no difference in the

behavior between larvae scooped with sucrose and larvae scooped directly from the food plate with a par of forceps.

The larvae were dried and spread on the agar starting from the center of the arena. The substrate for behavioral experiments was a 3% Bacto agar gel in a 25625 cm2 square plastic dishes. Larvae were washed with water at room temperature, the dishes were kept at room temperature and the temperature on the rig inside the enclosure was set to 25˚C.

The humidity in the room is monitored and held at 58%, with humidifiers (Humidifirst Mist Pac-5 Ultrasonic Humidifier).

We tested approximately 50–100 larvae at once in the behavioral assays. The temperature of the entire rig was kept at 25˚C. In the assay, the larvae were left to crawl freely on an agar plate for 44 seconds prior the stimulus delivery. The air-puff was delivered at the 45th second and applied for 38 seconds. After a period of recovery of 22 seconds when 10 air-puff pulses, 2 second each, were delivered (with a 8 second separation interval). We computed behavioral and transitional probabilities for different time-windows during stimulation. In this study, for the data from the behavioral screen, we show behavioral probabilities and transitions probabilities of the initial response of the larvae to the stimulus, immediately after the first stimulus onset. Specifically, we show the behavioral probabilities for the first second after the first stimulus onset and the transition probabilities for the first three seconds after stimulus onset.

The MWT software64 (http://sourceforge.net/projects/mwt) was used to record all behavioral responses.

## Screen design

We screened 567 GAL4 lines from the Rubin GAL4 collection (Jenett et al., 2012; Pfeiffer et al., 2008) were we silenced small subsets of neurons and individual neurons using tetanus toxin. We selected these lines from the entire collection for sparse expression in the brain and ventral nerve cord of the larval CNS as well as expression in the sensory neurons (images of the larval CNS are available at http://www.janelia.org/gal4-gen1). Out of the 567 lines tested, there were neuronal lines that were not part of the collection: we added 19–12 GAL4 and nompC-GAL4 for sensory neurons and OK107GAL4 for the mushroom body. We screened each GAL4 line in the air-puff assay described above.

## Behavioral analysis

**Larva tracking.** Larvae were tracked in real-time using the MWT software [55]. We rejected objects that were tracked for less than 5 seconds or moved less than one body length of the larva. For each larva MWT returns a contour, spine and center of mass as a function of time. From the MWT tracking data we computed the key parameters of larval motion, using specific choreography (part of the MWT software package) variables [31].

**Behavioral detection.** From the tracking data, we detected and quantified the behaviors: hunches, head-Bends (Bend), backwards crawls (Back-ups), stopping (Stop) and peristaltic crawling strides (Crawl) using a behavior classification that allows discriminations between all the different actions. Behavior classification was performed using incremental supervised learning based on human tagging of limited amount of larval videos i.e. [56]. It was also performed on a very limited set of features exhibiting low variance under the mechanical deformations induce by the larval dynamics. It was performed on incremental learning with new subsets of larva used for step of training. It consists on a 5-layer procedure with weak learners. The first layer relied on Random Forest [57] to identify if one of the listed behavior is being performed and output a Boolean variable. The second layer collects all states and check for inconsistencies (e.g. a larva crawling and bending at the same time). The third and fourth

layers used additional features and previous probability of actions (establish in previous layers) and Random Forest again to perform the behavior assessment [31]. Final layer combines features associated to behaviour estimation during their full duration to both random forest and Support Vector Machine for final classification. A graphic of the procedure is shown in S8A Fig.

**Tagging data.** Data tagging was performed using custom, simple GUI in Matlab. A first version provided the path of the larva, the contour and spine. The head, tail (non-differentiated) and neck points were apparent. The evolution of 2 features could be plotted to help tag some actions. Tagging was limited to whether an action was ongoing or not.

A second GUI (the 2 versions of it) with similar design displayed the state assigned by the previous layer of classification, the differentiated positions of the head and tail as well. One version was done with a local zoom on the larva to provide 2 simultaneous views, one more focused on global dynamics and one more local.

**Principle and method used to action detection.** The analysis is done in sequential layer using the following simple architecture:

- Loading and Cleaning input files

- Generate features time series

- Identify Head and Tail

- Generate first estimation of the actions

- Regularize these actions using imposed hierarchy and custom rules

- Generate second estimate of the actions

- Regularize singular events

- Action specific corrections

**Loading and cleaning input files.** Inputs of the analysis pipeline are made of the time series of the contours and the spine of individual larva as produced by the Choreography scheme [31]. Contours are dynamically generated live on subsets of larva from very different size and shape and thus have non-constant size. The average length is 114±23 points measures on a sample of 50 000 larvae. Spines, which are the skeleton of the larva, were generated offline and were all made of 11 points. Input files were then cleaned of the 5 times first points and last 10 time points which bear usually several anomalies (at the screen scale). We removed individual larva experiments that had not at least 250 points (lasted ~ 20s) and whose trajectories had a convex hull of surface inferior to 1mm$^2$ (nearly did not move at all).

**Generate features.** Complete details of all features computed are shown in the (https://github.com/DecBayComp/Pipeline_action_analysis_t5_pasteur_janelia)

Here, we list key features that provided selectivity in classification.

The center of the larva was defined as $\overrightarrow{r}_c = \langle \overrightarrow{r} \rangle_c$ with $\langle \rangle_c$ the averaging along the contour.

The "necks" of the larva were defined as $\overrightarrow{r}_{nd} = \frac{1}{2}\left( \overrightarrow{r}_{s3} + \overrightarrow{r}_{s4} \right)$, $\overrightarrow{r}_n = \overrightarrow{r}_{s6}$ and $\overrightarrow{r}_{nu} = \frac{1}{2}\left( \overrightarrow{r}_{s8} + \overrightarrow{r}_{s9} \right)$ where the index $nd$ stands for neck-down, $n$ for neck, $nu$ for neck-up and with $\overrightarrow{r}_{si}$ is the ith point of the spine.

We defined 4 segments in the larva, $S_{e1}$ joining the head to the neck-up, $S_{e2}$ joining the center to the neck up, $S_{e3}$ joining the neck-down to the center and $S_{e4}$ the segment joining the tail to the neck-down.

We defined $S = \frac{1}{2}(3\langle \cos^2\theta \rangle_s - 1)$ with $\langle \rangle_s$ the averaging along the spine curve, $\cos\theta$ the scalar product between normalized vector associated to a segment of the spine and the direction of the larva body define as the normalized vector joining the lower neck the mid point of the spine. S takes value between -0.5 and 1.

We defined 4 main angles as $\theta_{1-4}$ as respectively, $\theta_1$ the angles between $S_{e1}$ and $S_{e3}$, $\theta_2$ the angles between $S_{e4}$ and $S_{e2}$, $\theta_3$ the angles between $S_{e1}$ and $S_{e2}$, $\theta_4$ the angles between $S_{e3}$ and $S_{e4}$.

We defined $\lambda = \frac{\lambda_1 - \lambda_2}{\lambda_1 + \lambda_2}$, with $\lambda_i$ the eigenvalues of the matrix $\begin{pmatrix} \langle (x - \langle x \rangle)^2 \rangle_c & \langle (x - \langle x \rangle)(y - \langle y \rangle) \rangle_c \\ \langle (x - \langle x \rangle)(y - \langle y \rangle) \rangle_c & \langle (y - \langle y \rangle)^2 \rangle_c \end{pmatrix}$ with $\lambda_1 \geq \lambda_2$. $\lambda$ characterizes the shape of the larva and takes value between 0 and 1.

Various velocities or acceleration were evaluated both directly and then smoothed or by convolution with derivatives of Gaussian with different standard deviations. We evaluated the effective agitation velocity as $v_a = \frac{1}{3}\alpha_v(v_c^2 + v_{nu}^2 + v_{nu}^2)$ with $\alpha_v = 10$ for this screen analysis. We also evaluated the motion velocity and acceleration of the head, tail and center of mass of the larva.

We defined a set of linked variables defining direction of motion

$$\beta_{n_1, n_2} = \langle \overrightarrow{u} \rangle_{n1} \langle \overrightarrow{v} \rangle_{n2}$$

with $\langle \rangle_{n_i}$ the temporal Gaussian averaging over $n_i$ points, $\overrightarrow{u}$ the unit vector going from tail to neck down and $\overrightarrow{v}$ the unit vector in the direction of motion (center of mass).

We defined the various scale effective distance moved

$$d_{n_1, n_2} = \langle \| G'_{n1} * \overrightarrow{r}_{tail} \| \rangle_{n2}$$

with $G'$ the derivatives over $n_1$ point, $*$ the convolution, $\| \|$ the norm, and $\overrightarrow{r}_{tail}$ the position of the tail.

We defined the true and modified length of the larva as $l = \sum_{i=1}^{10} \| d\overrightarrow{r_{Si}} \|$ and $l_M = \sqrt{\sum_{i=1}^{10} \| d\overrightarrow{r_{Si}} \|^2}$ with $d\overrightarrow{r_{Si}} = \overrightarrow{r}_{Si+1} - \overrightarrow{r}_{Si}$.

We defined the effective rotation energies

$$\omega^{up/low} = d_{1/2}^{up/low}(\theta^{1/2})'^2$$

with $d_{1/2}^{up/low}$ the distance between head/tail and the center of the larva, $\theta'^2$ the squared derivative of the angle.

We defined the auxiliary direction variables as $\gamma_{k,l}^{\overrightarrow{w}} = \frac{(G'_{5} * \overrightarrow{r}_c).\overrightarrow{w}}{l}$ with $\overrightarrow{w} \epsilon \{\overrightarrow{u}_{k \to l}, \overrightarrow{v}_{k \to l}\}$, $\{k \to l\} \epsilon \{\text{neck} \to \text{head}, \text{neck} \to \text{tail}, \text{tail} \to \text{head}\}$ and $\overrightarrow{v}_{k \to l}$ the vector perpendicular to $\overrightarrow{u}_{k \to l}$.

Features were then smoothed at various time scales, derived, log-derived, and squared after derivation. Furthermore, minimal, maximal and cumulative values over time points window of various size were evaluated.

Some features were also dynamically normalized with respect to the length of the larva. Motion velocities were normalized by $v_l = \frac{l(i)}{dt(i)}$. The surface of the contour was split into two normalized variables, one normalized by the surface of convex hull of the contour the other by the surface of the circle of radius $r(i) = \frac{l(i)}{2}$. The direct distance from head to tail, the distance from the head to the neck and the neck to the tail were normalized by the length of the larva $l(i)$.

Finally, last layer classifications used global variables averaged over the full temporal duration of the larva dynamics: $\{\langle l \rangle_t, \langle \bar{v}_a \rangle_t, \langle \bar{v} \rangle_t, \langle \bar{d}_{5,5} \rangle\}$ where only the length of the larva is not normalized.

**Identify of head and tail.** In order to properly sample the large space of possible actions and to explore differences between individual larva, experiments were performed on large sets of larva. To reduce probability of contact between larva that would induced unwanted actions, agar plates were made large enough to have large portion of "free" motion during times or recording. In order to match all these conditions, the optical magnification was chosen to be low providing limited features on the larva. Thus, neither the mouth hook nor the difference of curvature between head and tail could be reliably detected. Hence, neither individual images nor contour shapes could be reliably used to identify head and tail.

Head and tail are the terminal point of the spine; hence both head and tail can only be in position 1 or 11. We relied on a Hidden Markov model (HMM) to detect head and tail from the larva dynamics. To reduce the dynamics to a one time step HMM, 4 hidden states were introduced corresponding to the 4 possible transitions between 2 time steps: $T_1$: (Head 1→1, Tail 11→11), $T_2$: (Head 1→11, Tail 11→1), $T_3$: (Head 11→11, Tail 1→1) and $T_4$: (Head 11→1, Tail 1→11).

The likelihood model was defined to reinforce global characteristics of head and tail dynamics when recorded in high resolution. The likelihood at the time point, i, reads:

$$P_i(T_j) \propto exp(-\beta(\varepsilon_c(i, T_j) + \varepsilon_{d1}(i, T_j) + \varepsilon_{d2}(i, T_j) + \varepsilon_a(i, T_j) + \varepsilon_v(i, T_j)))$$

with

$$\varepsilon_c(i, T_j) = \alpha_c(\bar{\bar{\xi}}(j, 1)\delta\kappa(i) + \bar{\bar{\xi}}(j, 2)\delta\kappa(i + 1))$$

$$\varepsilon_{d1}(i, T_j) = \alpha_{d1}(\bar{\bar{\xi}}(j, 1)\beta_{10,10}(i) + \bar{\bar{\xi}}(j, 2)\beta_{10,10}(i + 1))$$

$$\varepsilon_{d2}(i, T_j) = \alpha_{d2}\bar{\bar{D}}(T_j, i)$$

$$\varepsilon_a(i, T_j) = \alpha_a(\bar{\bar{\xi}}(j, 1)a(i) + \bar{\bar{\xi}}(j, 2)a(i + 1))$$

$$\varepsilon_v(i, T_j) = \alpha_V(\bar{\bar{\xi}}(j, 1)v_c(i) + \bar{\bar{\xi}}(j, 2)v_c(i + 1))$$

with the matrix [58], $\delta\kappa = \tanh(\kappa_S(1) - \kappa_S(11))$ with $\kappa$ is in radians and defined in [58],[58].

In order to regularize the contour of the larva and evaluate the curvature at the Head/Tail points we projected and reconstituted the larva contour onto a Fourier basis. This procedure was directly inspired by a step of the closed-loop analysis code found in [10,31]. We limited the decomposition and recomposition to the order 7 in coefficients. Curvatures were measured using the reconstituted (smooth) larva contour.

Parameters were set to $\alpha = (1, 0.2, 1, 0.5, 0.5)$, $\beta = 0.01$, $p_{init} = \left(\frac{1}{4}, \frac{1}{4}, \frac{1}{4}, \frac{1}{4}\right)$ with a transition matrix $\bar{\bar{T}} = \begin{pmatrix} \bar{\bar{A}} & 0 \\ 0 & \bar{\bar{B}} \end{pmatrix}$ with $\bar{\bar{A}} = \bar{\bar{B}} = \begin{pmatrix} 0.9 & 0.1 \\ 0.1 & 0.9 \end{pmatrix}$. The transition matrix was purposefully set to much higher transition values than temporally observed. It allowed possible changes following complex maneuvers. Note that the same value was used at the screen scale. Another analysis could be performed by adapting the transition matrix to the shape and dynamics of the larvae.

The head and tail position were derived from the Viberti path (path of hidden states maximizing the likelihood). As for all HMM process the longer the trajectory the better the results.

The main source of error stemmed from the transition from ball form to a moving dynamic. When the larva is curled the spine is constantly moving due to the lack of high curvature area to stabilize it. If the larva performs a ball shape moves for a short time and then performs a ball again, there are limited constrain to define head and tails.

Notably, only for the crawling backward is the head and tail definition effectively used.

**The multilayer architecture.** We point out here the reason for the incremental multiple layers of classification. Behavior or action tagging in larva by human observers is not fully reliable. Different people tend to see different dynamics when looking at the contour and time series dynamics. For example, different researchers have implicit "amplitude based"limits of behavior, *i.e.* two experts may diverge on a behavior being performed based on assumptions regarding amplitudes. For example, can a hunch with a small amplitude (less than 5% difference in larva length) be considered as a hunch or can the lateral small head bends (less than 10˚) not associated with motion direction changes be considered as bending? Finally, in many cases the low resolution of the contour makes the choice of dynamics unsure. Hence, first layers of classification were performed on limited larva exhibiting clear dynamics and any unsure actions were not tagged. These actions were then used to classify actions at the full screen scale. This allowed future tagging to be performed with the first layer's state being displayed. This process helped to unify description of the dynamics and provided clearer elements to judge the action dynamics.

**Generate first estimation of the actions: First layer classification.** In order to reduce the effect of misclassification of the head and tail, all classification except the ones explicitly for crawling backward were done without requiring their detections.

Classifications were performed using Random Forest on the small samples of tagged larva. Selected sets of features were provided in relation to the action to be tagged. Action detection was performed independently for each action with results being at each time point the action being on or off. The 5 actions and additional actions were detected.

Additional postural states were also detected. These postural states simply describe nondynamic the shape of the contour. They were divided into 4 non-overlapping state: straight, Bend, curl and ball. An additional fifth state straight and light Bend was added to improve backward motion detection especially when associated in sequences involving Bend and backup.

Crawling backward was defined as the combination of the straight and light bent (Bend) postural state, the displacement status and the feature axis_direction_25_10 taking values inferior to the threshold -0.8.

**Regularized actions using imposed hierarchy and custom rules.** The first layer of tagging was performed on limited set of larvae. Hence, we left at this stage of the code, the possibility of having a state with no action classification. Furthermore, each action was defined independently of the other. This no action set was defined hierarchically transferring

- Any action happening at the same time as stop was set to no-action

- Any action happening at the same time as roll was set to no-action

- Any action happening at the same time as back-up was set to no-action

- Any action happening at the same time as Bend was set to no-action

- Any action happening at the same time as crawl was set to no-action

Similarly, postural states were regularized hierarchically, using the probability associated to the Random Forest as defining criteria. The postures were regularized sequentially in the following order ball→curl→Bend→straight.

**Generate second estimate of the actions: Second layer classification.**   The second layer classification was based on Random Forest. All actions were trained at the same time with an output being of the 6 actions. All actions were tagged and the no-action states were removed. Finally, to the time series of features computed on the larva contour and spine, the actions classification from the previous layer and their smooth probabilities were added as an input of the random forest. Auxiliary actions and states were not re-trained on this layer.

**Regularize singular events.**   The nature of the larva dynamics and the scale (temporal and spatial) at which the behavior is analyzed prevent certain transition to be possible. We thus regularized two patterns: i→j→i and i→j→k where {i,j,k} are one of the possible actions, and where these actions are separated by 1 time point. The i→j→i pattern was systematically corrected to i→i→i and the i→j→k pattern to i→k→k.

**Action specific corrections.**   Action specific corrections are based on evaluating parameters during the action as defined in the previous layers.

**Hunch.**   New features were generated to ensure a better discrimination when the action head retraction was detected. These features were associated to the length of the larva and to the average, minimum and maximum values taken during the detect head retraction action. A random forest was the train to differentiate between head retraction and other quick motions that could be taken as head retraction.

**Stop.**   Stops lasting less than 2 time points were transformed into the closest action they could be performing.

**Back-ups.**   Back-ups lasting less than 2 time points were transformed. This correction was implemented to lift the ambiguity of nearly stopped larva where small back mostly consequences of the resolution of the tracker were generated.

**Action continuum vs discrete actions.**   Larva behavior was analyzed at the screen scale under the discrete actions hypothesis. Continuous actions vs discrete actions remain a partially ill posed problem as that nature of behavior encodings in the nervous system remain inaccessible. Behaviors are only accessible (in this paper) through their muscular outputs. Hence, discrete vs continuous actions are associated to discrete clustering vs continuous representation of larva-associated features, which is also an ill-posed problem. Our analysis was supported by natural clustering of larva actions using the features designed to robustly capture the motion. In S1 Fig. we showed projection of 3 parameters associated to the larva. Natural clusters appear corresponding to 6 main actions performed by the larva. Coloring was performed using the pipeline. The 3 parameters used $x_1 = log\left((1 - \langle S \rangle_{10})^2\right)$, $x_2 = \frac{\langle v \rangle_{10}}{\langle l \rangle_{25}} \alpha_{10,5}$, $x_4 = \sqrt{\langle \left(\frac{dl}{dt}\right)^2 \rangle_{10}}$, with $\alpha_{10,5} = \overrightarrow{\langle V \rangle_{10}} \cdot \overrightarrow{\langle U \rangle_5}$, where $\overrightarrow{U}$ is the unit vector going from the tail to the mid point of the spine and $\overrightarrow{V}$ is the unit vector in the direction of motion of the mid-point of the spine (the direction of motion). Each point in the plots is the average value of the parameters in one experiment when the larva performs on the behavior. We also show similar plots on example of hits in the paper. Note that for all, the parameters associated to actions are stable ensuring stability of behavior representation with the various lines.

**Heuristic to automatically characterize action detection quality.**   The size of the screen prevents manual checking of all experiments in order to ensure stability of action and behavior representation. We designed a simple scheme to get a global sense of the quality of the description by the pipeline. For all lines, we extracted the average values of a limited set of features (14) associated to all behavior and trained classifiers to predict behavioral output using

supervised learning (with random forest) and unsupervised learning (k-nearest neighbors). We evaluated the efficiency of the scheme using cross validation and outputted the accuracy for all individual behavior (S8B Fig)

This scheme primarily shows that the pipeline tends not to overfit as a limited set of parameters can predict behavior. Empirically, it was found that high score was associated to non-ambiguous while lower score would be associated to more ambiguous action that would not be easily characterize by an expert viewer.

**Action definition.**   **Stop**. A larva is stopped if it does not show any detectable muscular activity. A non-crawling larva is stopped if it does not dynamically bend. A stopped larva can be in a bent posture.

**Hunch**: A larva is hunches or Head-retracting if it is dynamically (and most of the time more quickly than the rest of the larva dynamics) diminishing its length beyond peristaltic wave amplitude. Head retraction is not associated to a backward movement.

**Bend**. Any dynamics of bending in the larva whether it is displacing (turning movements) or not displacing (head or tail Bending movements). Here, displacing means the center of mass or the center of the larva (not necessarily the main point) is moving.

**Crawl**: Larva displacement in the direction going from the tail to the head.

**Back-up**: Larva displacement in the direction going from the head to the tail.

**Rolling**: A larva is rolling it is moving laterally while being bent. Rolling is not present in this study (Roll is a type of nociceptive response, that doesn't normally occur in response to air-puffs.

**Choice in data used for training.**   The Behavioral larva database is made of times series of larva contour and larva spine. All are untagged. Probability of actions in drosophila larva is extremely heterogeneous and some actions can only be seen following specific stimuli patterns. For example, a sample of nearly 14000 larvae taken in the reference (w;;attP2) exhibit the proportion of events follow the following raw statistics (with small motions associated to their closes behavior): 50% crawling, 37% Bending, 9% stopped, 1% head retracting, 3% crawling backwards (Back-ups). Hence, in order to properly sample a large variety of events tagging was done selectively with a focus on pre and post stimuli (the main at 45 s but also the series of 10 stimuli beginning at 95s). Data for training came from multiple experiments with similar focus on activation time. Initial tagging was performed on "hits"and reference lines described in [10,31]. Then, as the behavior detection scheme was designed using incremental learning, new batches of larvae were tagged at each step. At each stage larvae were selected in lines exhibiting either low or high probability of detecting actions with the adjunction of few larvae randomly selected.

**Temporal anomaly.**   Note that a dependency exists in the MWT system between the number of larvae actively tracked and the time between frames. The average time between frames is $dt = 8.2 \ 10^{-2} \pm 1.3 \ 10^{-2}$ s with minimal value of $dt_{min} = 2.8 \ 10^{-2}$ s and maximal value of. $dt_{max} = 2.9$ s. These values were evaluated on a sample of 50000 larvae and total corresponding recording time of $T = 56000$ s randomly sampled from the full experimental database. Local variability of dt may degrade the quality of behaviour assessment.

**Anomalous locomotor defect phenotypes.**   We defined anomalous phenotypes at the population scale. Non-stimulated larva moving freely on agar plates exhibits two actions: forward crawling with an average time of ~10s and Bending with an average time of ~1s (with occasionally random stops and crawling backward). These two dynamics follow roughly a Poisson dynamics. Hence at any time, at the population scale crawling is the dominant behavior (the main behavior happening) and the second dominant is Bending. Hence,

$H_0$: the normal locomotors behavior if P1m>Ptresh and P2m>Ptresh

$H_a$: the abonormal behavior if P1m<Ptresh or P2m<Ptresh

With m = 567, k = 1:crawling, k = 2:bending, Ptresh = 0.97, Pkm = <I(k,t)>T with I the indicator function with

I(k,t) = 1, when the larva is performing action k during the time [t,t+dt] and I(k,t) = 0 otherwise $\langle\rangle$T is the temporal average during time window T. The time window T = 44.75s was set to provide 250ms safety prior to the stimuli. All lines in $H_a$ we removed experimental sets if this hierarchy was violated prior to the stimuli at 45s. They are referred in the main text as lines most likely exhibiting locomotor defects.

**Evolution of action definition.**   In this paper, action definition was generalized from the ones in [10,31] while remaining compatible. The goal of current discrete definition was to remove "amplitude"parameters in order to be able to associate to each action, a nature (running bending etc), a duration (from start to stop), amplitudes (for example velocity of the run, angular amplitude of bending, length change during hunching etc.) and two transitions from and to another action. We provide here, links between former and current definitions.

**Head retraction vs hunch.**   In previous studies [31], "hunch" was defined as the strong length decrease of the larva with the thresholds as describe in [10]. "Hunch", as defined in the present study, are all events of larval shortening regardless of their magnitude. If we apply a similar threshold (effective length change dl = 0.8) used for previous "hunch" detection with the present machine-learning behavioral detection method we obtain previously defined hunches.

**Bend vs bend.**   Similarly, previously [31] "bends" were described as events with head-tail angles of 27˚ or more [59]. If we apply a similar threshold to the "Bend" events detected with the present machine-learning algorithms, we obtain the previously defined "bends" (effective angle change, S = 0.81).

## Quantification and statistical analysis

**Ethograms.**   Ethograms were designed to display the temporal evolution of the individual larva behavior with time. Color code is associated to the action being performed. Stimuli are displayed in magenta thick lines.

**Behavior probability.**   The temporal evolution of behavior was computed as the ratio of the number of larva performing the action over the total number of active larva in time windows of duration 2dt with dt = 100ms. Noticeable evolution of noise in the curves may be the result of varying number of active larvae in the rig.

**Dominant behavior.**   The temporal evolution of dominant and second dominant behavior was computed as the most probable and second most probable behavior in same time window as behavior probability.

**Transition analysis.**   Transition analysis simply consisted of counting during a predefined time window all the transition between behaviors. These transitions are represented in matrices where initial states are rows and arrival states are columns. Normalization was performed on rows. The transition probabilities starting from a given behavior for control and experimental populations were statistically compared by the Fisher exact test. We say that any p-value less than 0.05 (uncorrected for multiple comparisons) is significant.

**Sequence analysis.**   Sequences were measured as the list of different actions performed after the stimulus. We limited the number of actions to the first four actions performed after stimulus onset. We defined $dt_{blur}$ = 500ms as the time of uncertainty if the stimulus has been perceived. If the larva performs the same actions as it was before the stimulus for longer than $dt_{blur}$ it is considered to be the response to the stimuli. We included in this analysis all the actions: hunches, Bends, back-ups, stops as well as the baseline behavior crawl.

We used a chi-squared statistical test to compare GAL4 lines with control (w;;attP2). We say that any p-value less than 0.05 (uncorrected for multiple comparisons) is significant.

**Behavioral probability analysis (time windows).**   Probability analysis was performed on averaging on various time windows, 1s, 5s and 15s, the fraction of time that the larvae spent performing an action. This analysis was performed after the start and the end of the stimuli. We used a chi-squared statistical test to compare GAL4 lines with control (w;;attP2). We say that any p-value less than 0.05 (uncorrected for multiple comparisons) is significant.

**Cumulative analysis.**   Cumulative analysis was performed by counting the number of larva performing at least once an action in various time widows of duration 1s, 5s, and 15s. This analysis was performed after the start and the end of the stimuli.

**Larva length analysis.**   Length analysis was performed during head retraction event. We evaluated the maximal and minimal length (both $l$ and $l_M$) during the hunch and measured the relative variation $\delta(l) = \frac{l_{max}-l_{min}}{l_{max}}$ and $r(l) = \frac{l_{min}}{l_{max}}$.

**Amplitude during bend.**   The amplitude of Bend was analyzed during Bend event after the stimulus. We estimated the average value of the minimal S value during the Bend.

**Statistical significance and hits preselection.**   The goal of the paper is to identify possible hit neurons involved in significant behavior probability modification as well as those involve in competitive interactions between mutually exclusive actions. Hence, we design a screen in order to select a list of possible hits from the behavioral output.

**1)** Prior to any analysis, we investigated if lines did not exhibit anomalous behavior (see Anomalous locomotors defect phenotypes) excluded all lines exhibiting locomotors defects (see S2 Table) prior to any stimulation. Hence, from the initial 567 lines, 95 lines were removed leading to a subset of 471 lines + 1 control to analyze.

**2)** Analysis was performed as planned comparison where all lines were compared to the control (w; attP2-TNT). First we seek for lines that might induce increase/decrease of behavior probability after the stimulus when compared to the control. Hence,

$H_0$: the null hypothesis $\pi_m^k = \pi_0^k$

$H_A$: the hit hypothesis $\pi_m^k \neq \pi_0^k$

with $k \in [1..7]$ 1:crawling, 2:bending, 3:stopped, 4: hunching, 5:carwling backward, 6: rolling (not evoked here) 7: small motions, $m \in [1..471]$ the lines being tested and m = 0 the control.

The distribution of $p(\pi_m^k) = B(\pi_m^k, N_m, n_m^k) = \binom{N_m}{n_m^k}(\pi_m^k)^{n_m^k}(1 - \pi_m^k)^{N_m - n_m^k}$ is the binomial distribution. We performed the generalized likelihood ratio test statistic between each lines and the control with $z = -2\log\left(\frac{B(\pi^k, N_m, n_m^k)B(\pi^k, N_0, n_0^k)}{B(\pi_m^k, N_m, n_m^k)B(\pi_0^k, N_0, n_0^k)}\right)$ with $\pi = \frac{n_m^k + n_0^k}{N_m + N_0}$ and $p = \chi^2(z, df = 1)$ as well as Chi squared test (which is asymptotic limit of the likelihood ratio test).

**3)** Simultaneously, we tested for significant difference between lines regarding the full distribution of behavior, with

$H_0$: the null hypothesis $[\pi_m^1, .., \pi_m^7] = [\pi_0^1, .., \pi_0^7]$

$H_A$: the hit hypothesis $[\pi_m^1, .., \pi_m^7] \neq [\pi_0^1, .., \pi_0^7]$

The distribution of $p([\pi_m^1, .., \pi_m^7]) = Mult(\Pi_m, N_m) = \frac{N_m!}{\prod_{k=1}^{7} n_m^k!}\prod_{k=1}^{7}(\pi_m^k)^{n_m^k}$ is the multinomial distribution with $N_m = \sum_{k=1}^{7} n_m^k$. Similarly we performed both the generalized likelihood

ratio test statistic with $z = -2\log\left(\frac{Mult(\Pi,N_m)Mult(\Pi,N_0)}{Mult(\Pi_m,N_m)Mult(\Pi_{m0},N_0)}\right)$ with $\Pi = \left[.., \frac{n_m^k + n_0^k}{N_m + N_0}, ..\right]$ $p = \chi^2(z, df = 6)$ and the Chi squared test.

**4)** We also performed planned comparison for possible competitive interaction lines. We used two simultaneous tests to search for candidate hit lines. We first used the test **3)** for detection of hit lines regarding full distribution of behavior. The test was complemented for the sake of selectiveness by the following test

$H_0$: the null hypothesis $(\pi_m^{k_1} = \pi_0^{k_1})\&(\pi_m^{k_2} = \pi_0^{k_2})$

$H_A$: the hit hypothesis $(\pi_m^{k_1} \neq \pi_0^{k_1})\&(\pi_m^{k_2} \neq \pi_0^{k_2})$ with $((\pi_m^{k_1} > \pi_0^{k_1})\&(\pi_m^{k_2} < \pi_0^{k_2}))\|((\pi_m^{k_1} < \pi_0^{k_1})\&$ $(\pi_m^{k_2} > \pi_0^{k_2}))$

with $k_1 \neq k_2$, $(k_1, k_2) \in [1,..7]$, where each behavior is considered to be statistically independent of the other. We used results from **2)** and required for one of the two behaviour to have the p-value inferior to $p_{tresh} = 0.01$ and the second one to at least be inferior to $10.p_{tresh}$.

**5)** Independently, we sought hits leading to large deviation in amplitude associated to behavior from the control. We investigating changes in length variation for hunch, the parameter S (acting as an effective angle) variation for bending and the velocity of motion variation for forward crawling. Hence,

$H_0$: the null hypothesis $A_m^k = A_0^k$

$H_A$: the hit hypothesis $A_m^k \neq A_0^k$

With A the associated amplitudes, $k \in \{2,4,1\}$ and $m \in [1..471]$ the lines being tested and m = 0 the control. We used Two-sample Kolmogorov-Smirnov test to compare amplitude.

**6)** On the subset of selected hit lines, we investigated the transition matrices. Analysis was performed as planned comparison, on the subset, where all lines were compared to the control (w; attP2-TNT). Transitions matrices elements were defined as $T_{kl} = \frac{N_{k \to l}}{\sum_l N_{k \to l}}$ with $(k,l) \in [1..7]$, $N_{k \to l}$ the number of events where a larva goes from the behavior k to the behavior l. The behavior being defined as mutually exusive, we have $N_{k \to k} = 0$ and $\forall k \in \{1..7\}$ $\Sigma_l T_{kl} = 1$. We sought for lines that might induce increase/decrease in transition probability when compared to the same transition probability in the control. Hence,

$H_0$: the null hypothesis $(T_{kl})_m = T_{kl0}$

$H_A$: the hit hypothesis $(T_{kl})_m \neq T_{kl0}$

Similarly, the distribution of $T_{kl}$ is binomial distribution and we use generalized likelihood ratio test statistic and the Chi Square test for statistical testing (S2 Data file, S4 Data file).

We further used the same approach to investigate transition matrices screenscale (S11 Table, S5 Data file)

Note that initial assessment of possible hits worth investigating, whether for pure change of probability or coupled simultaneous increase and decrease of probability, was done following previous successful approach in finding neuron lines in [59], by using directly the p-value of the various tests performed against the control, thus not corrected for multiple comparison. In doing so, we increased the risk of false positive detections. The most general correction for the multiple testing is the Bonferroni correction where the criteria $\alpha^* = \frac{\alpha}{m}$, with here m = 471 (1 test vs control per line).

We chose different level of significance regarding the competitive interaction neuron hit list (simultaneous increase and decrease of probability of a behavior) and the neuron list for

simultaneous increase/decrease of multiple behaviors. It was a pure commodity to reduce list of possible hits and focus on those whose had sparse expression patterns or those connected to some neurons already known to have a key role in decision-making.

Note that hits were detected involving selectively the behavior small but were not discussed in this paper. Small motion gathers all behaviors characterized by small amplitude such as Hunches with limited relative length variation, crawling with small motion velocities or bending with small S variations etc. Diversity of conditions of small motion behavior emergence suggests this behavior may exist as a separate category of actions in the larva nervous system. Yet, further Investigation of this behavior will require amongst other things large-scale recordings at higher resolution to ensure a better characterization of their associated features.

**False negative and false positive.**   As stated before, the list of hits that we identify provide candidate neurons whose involvement in in sensorimotor decisions needs to be confirmed by combining phenotype information with connectome information and functional studies. Thus, although a large statistical significance between a neuron line compared to the reference remain a strong indicator of significant involvement of this neuron only with supplementary information outside the screening procedure can we confirm their involvements. Conversely, being not being selected as hit due to lack of statistical significance for individual test or for multiple testing corrected test does not exclude the neuron from having a significant role.

**Behavioral response dependency on larva angle at the start of the air puff.**   In order to question whether the air puff angle did play a role in behavior selection, we computed the cumulative distribution function (CDF) of the larva angles (for the control: w;;attP2-TNT larvae) when compared to the wind for all behavior when the stimulus started. Results are shown in S9 Fig. CDF curves did not exhibit specific patterns (safe from stopping action) suggesting preferred angle for decision making when related to the wind direction. We perform KS test to compare the main actions considered in this paper with: $p = 0.2$ for Hunch-bend, $p = 0.18$ for bend-back and $p = 0.04$ for hunch-back.

## Supporting information

**S1 Fig. Natural clustering of behaviour in Drosophila larva. A.** Scatter plots of 3 larva features with 3 projections for the control and 10 hit lines (plotted per behaviour). Each point is the average value of a larva feature when performing one of the behaviour during one experiment. Colour code is associated to behaviour with: in black Crawl, in red Bend, in Green Stopped, in blue Hunching, in Cyan Back-up and in grey-small motions. The plot was performed on ca 13000 larvae from the w::attP2-TNT experiments. The features shown here are

$x_1 = log\big((1 - \langle S \rangle_{10})^2\big)$, $x_3 = \frac{\langle v \rangle_{10}}{\langle l \rangle_{25}} \alpha_{10,5}$ and $x_4 = \sqrt{\langle \big(\frac{dl}{dt}\big)^2 \rangle_{10}}$. Note the natural separation of the features with the behaviour; small motion lies in between Crawl, Bend and Stop. Hunch and Back-up are close together. You can also note the difference in the geometry of the features linked to action. B. Same as in A for the control plotted per larva **C**. Accuracy of behavior identification with limited features with in knn classifier (*top panel*) and in random forest classifier (*bottom panel*)
(TIF)

**S2 Fig. Additional characterization of the behavioral response of wild-type larvae to an air-puff. A.** Behavioral probabilities upon stimulus onset (during the 1[st] second of stimulation at high (6 m/s) and low (3 m/s) intensities of air-puff. p-values are all p<0.001 (\*\*\*). **B.** Scatterplots of Amplitude of Bending (effective angle) vs Bend probability. **C.** Scatterplots of Amplitude of Hunching (Relative length variations) and Hunch probability. **B-C.** Each dot

represents the average value for a neuronal line.
(TIF)

**S3 Fig. Inter-trial and inter-individual variability of mechanosensory responses.** A-F. Ethogram of the behaviors of the attP2>TNT control before, and upon repeated presentation of 2 s air-puff stimulation. Each line is a larva. Different colors represent different actions: Blue-Hunch, Red-Bend, Cyan-Back-up, Black-Crawl and in Green-Stop **A-E.** Different examples of inter-trial variability (within each ethogram) and inter-individual variability (between different ethograms). **A**. Example of larvae that hunched on the first stimulus, bended on the second stimulus and performed any action on the third stimulus. **B**. Example of larvae that hunched on the first stimulus, performed a small action on the second stimulus and performed any action on a third stimulus. **C**. Example of larvae that bended on the first stimulus, hunched on the second stimulus and performed any action on the third stimulus **D**. Example of larvae that backed-up on the first stimulus, and performed any action on the second and third stimulus **E**. Example of larvae that performed a small action on the first stimulus, hunched on the second stimulus and performed any action on a third stimulus **F.** Example of larvae that only hunched immediately upon repeated presentation of an air-puff stimulus
(TIF)

**S4 Fig. Behavioral probabilities for all five actions.** A-F Behavioral probabilities upon stimulus onset (during 1$^{st}$ second) for all the selected lines shown in Fig 4. for the five behaviors: Crawl, Bend, Stop, Hunch and Back-up are shown l *:p<0.05, **:p<0.01, *** p<0.001 **A**. R11A07, **B.** R39H05, **C**. R88C11, **D**. R66A12, **E**. R13A08, **F**. R32E04.
(TIF)

**S5 Fig. Ethograms of competitive interaction hit lines.** Ethograms for all the selected competitive interaction hit lines with sparse neuronal expression patterns shown in Fig 6.
(TIF)

**S6 Fig. Putative mechanosensory network.** –a summary diagram of all the neurons identified in the hit lines as putative neurons involved in mechanosensory responses in this study and some previously published key neurons involved in mechanosensory responses: Basin-1-4 (B1-B4), LnA, LnB, fbLN, A09e [10,21,38]. (Their detailed connectivity can be found in [10,21,30,38]). Connectivity is based on previously published reconstruction and newly reconstructed connections (S1 Data file, connectivity matrix). The arrows indicate excitatory connections, T-bars indicate inhibitory connections and plain lines when it is unknown whether it is excitatory or inhibitory. The width of the line is proportional to the strength of the connection. Some weak connections are not shown for clarity the number of synapses may be underestimated due to the lack of data from segments. Putative connections are shown as dashed lines. In addition to the reconstructed network in the VNC, putative neurons in the SEZ and brain are shown based on the candidate neuron from the behavioral screen (i.e. lines R92H01, R11F06, R82B06, R65B04, R27B12)
(TIF)

**S7 Fig. Putative mechanosensory network reconstructed in the CNS volume.** All the elements from the putatitive mechanosensy network shown in S6 Fig. Dorsal view (left panel), Lateral view (right panel).
(TIF)

**S8 Fig. A.** Graphical representation of the learning procedure. **B.** Accuracy of behavior identification with limited features with on the left, knn classifier and on the right, random forest classifier. Every line is a GAL4 line and every column is an action: Fc is Forward Crawl, B-u is

Back-up and Sm is Small motion. Note the overall constant accuracy throughout the lines.
(TIF)

**S9 Fig. Behavioral responses as a function of larval orientation at the onset of air puff.** Evolution of the Cumulative Distribution Function (CDF) of the angle $\theta = \theta_{\overrightarrow{e_1}} - \theta_w$ with. $\theta_{\overrightarrow{e_1}}$ the absolute angle of the $\overrightarrow{e_1}$ vector (top neck to head of the larva) and $\theta_w = 3\pi/2$ is the angle associated to the wind direction, with the first action after the air puff stimulus. Color code is associated with behavior (similar to all other plots in this paper) with small actions in purple. Note that linear evolution of the CDF is associated to flat (no preferred direction) of the probability density function (pdf).
(TIF)

**S1 Table. Probabilities screenscale_timewindow1.**
(XLSX)

**S2 Table. Lines with Locomotor defects.**
(XLSX)

**S3 Table. Amplitude Bend.**
(XLSX)

**S4 Table. Amplitude Hunch.**
(XLSX)

**S5 Table. Sensory line hits.** Behavioral probabilities for all five actions. Source data For Fig 3E
(XLSX)

**S6 Table. CNS line hits.** Behavioral probabilities for all five actions. Source data for Fig 3F and 3G.
(XLSX)

**S7 Table. CNS line hits.** Behavioral probabilities for lines with decreased (24 lines) or increased probabilities (1 line) in multiple actions.
(XLSX)

**S8 Table. Anatomical annotations.**
(XLSX)

**S9 Table. List of neurons.**
(XLSX)

**S10 Table. Probabilities_competitive interaction lines.**
(XLSX)

**S11 Table. Transition probabilities.** Transition probabilities for selected hit lines. For each transition probability, a p-value for comparison with the corresponding transition probability in the control (w;; attp2>TNT) are also shown.
(XLSX)

**S1 Data. Connectivity matrix.**
(CSV)

**S2 Data. Spreadsheet pvalues_screenscale.**
(XLSX)

**S3 Data. Spreadsheet screened lines.**
(XLSX)

**S4 Data. Spreadsheet pvalues for behavioral probabilities comparisons generalized likelihood ratio test.**
(CSV)

**S5 Data. Spreadsheet pvalues for transition probabilities, generalized likelihood ratio test.**
(CSV)

## Acknowledgments

We thank C.Montell (UC Santa Barbara), YN Jan (UC San Francisco) for fly stocks. We thank Fly core at JRC for fly crosses and Rebecca Arruda and Tam Dang for help with the behavioral experiments, Jim Truman for flp-out images of selected hit lines, Casey Schneider-Mizell for helpful discussion on the EM reconstruction and EM reconstruction of the hook-1 neuron) Michael Winding for helpful discussion on matching LM and EM images. This work used the computational and storage services (TARS cluster) provided by the IT department at Institut Pasteur, Paris.

## Author Contributions

**Conceptualization:** Jean-Baptiste Masson, Marta Zlatic, Tihana Jovanic.

**Data curation:** Tihana Jovanic.

**Formal analysis:** Jean-Baptiste Masson, Tihana Jovanic.

**Investigation:** Tihana Jovanic.

**Methodology:** Jean-Baptiste Masson.

**Project administration:** Marta Zlatic.

**Resources:** Albert Cardona.

**Software:** Jean-Baptiste Masson, François Laurent, Albert Cardona.

**Supervision:** Marta Zlatic, Tihana Jovanic.

**Validation:** Chloé Barré, Nicolas Skatchkovsky.

**Visualization:** François Laurent, Tihana Jovanic.

**Writing – original draft:** Tihana Jovanic.

**Writing – review & editing:** Jean-Baptiste Masson, Marta Zlatic, Tihana Jovanic.

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
