## [Decision Letter · Decision Letter 0]

3 Nov 2019

Dear Dr Jovanic,

Thank you very much for submitting your Research Article entitled 'Identifying neural substrates of competitive interactions and sequence transitions during mechanosensory responses in Drosophila' to PLOS Genetics. Your manuscript was fully evaluated at the editorial level and by independent peer reviewers. The reviewers appreciated the attention to an important topic but identified some aspects of the manuscript that should be improved.

We therefore ask you to modify the manuscript according to the review recommendations before we can consider your manuscript for acceptance. Your revisions should address the specific points made by each reviewer.

[LINK]

Yours sincerely,

Aravinthan Samuel

Guest Editor

PLOS Genetics

Gregory Barsh

Editor-in-Chief

PLOS Genetics

Reviewer's Responses to Questions

**Comments to the Authors:**

Reviewer #1: The review is uploaded as an attachment.

Reviewer #2: Review of Masson et el PGENETICS-D-19-01524

Masson et al describe the results of a circuit screen that was carried out using larvae of Drosophila melanogaster. 567 Janelia GAL4 lines were pre-selected for anatomically sparse expression patterns in the larval brain and/or peripheral nervous system. These lines were then crossed to a UAS-TnT to silence the neurons and larvae with the silenced neurons were exposed to an air puff stimulus. Video recordings were made of the animals with the silenced neurons before, during, and after responding to the stimuli, and the behavioral responses were analyzed using a supervised machine learning approach. Five general categories of behavioral responses are seen in various probabilistic combinations in response to an air puff stimulus (hunch, crawl, bend, stop, backup). By comparing the responses of the silenced animals to controls, the authors identified GAL4 lines that resulted in alterations in behavioral probabilities. Neurons targeted by these lines were then described and investigated further. Neurons identified were in the peripheral nervous system and distributed throughout the brain. The authors conclude that the results are consistent with a “distributed model of decision making.” The paper is very rich in data, and the experiments are well-performed, and the study will be of interest to the readers of PLoS Genetics. However, the conclusion that the data support a distributed model of “decision making” is overstated and this needs revision and further discussion

Major Points:

1.) The methods contain a very detailed discussion of the mathematical parameters used in the behavioral classification scheme, but there is almost no discussion of this in the text. The manuscript would be improved by a short discussion on the types of errors that can arise from this analysis. Are there caveats that the readers should be made aware of?

2.) The authors should justify their use of the term decision making. Although the term “decision making” is thrown around broadly in the field of biology, when it is used in the context of animal behavior and ethology there is a very specific meaning. In the context of behavior, the term “decision making” should be reserved specifically for when the organism is making choices between options that have quantifiable costs and benefits.

It is true that the term “decision” is more loosely used in the field of neuroscience in other contexts. For instance, during development, one can say that an axon makes a decision to either be repelled or attracted in a morphogen gradient. It can also be said that an enhancer “makes a decision” to be activated or repressed depending on the constellation of transcription factors that are present in a cell. A neuron makes a decision to fire or be repressed from firing depending on the balance of excitatory and inhibitory inputs that it receives.

But using the term decision making in the field of neuroethology necessitates that there is utility in the choice. A decision involves weighing costs and weighing the quality of the information that is being used in assigning values to the choices. These ideas are embodied and clear with valid uses of the term. A fly makes a decision when it is either attracted or repelled by an odor depending on whether or not this odor has been previously paired with an appetitive or an aversive stimulus. Or an odor can be attractive or repellent based on innately (evolutionarily) determined costs and benefits. A fly decides to lay an egg or to hold an egg depending on the complexities of the sensory inputs at the oviposition sites. The energetic resources involved with producing the egg are a clear cost to making an incorrect decision.

Thus, it can be argued can whether or not the larval behaviors under investigation involve genuine decision making. What is the cost to the larvae of producing one behavior or another in response to the air puff? What is the benefit? While it is clear that the neuronal network in the larval brain will produce a variety of behavioral outcomes if given an apparently identical stimulus, it is less clear that this is the result of a decision. The probabilistic nature of the response makes it more akin to the rolling of dice than to making a decision.

It is likely that the responses may have utility with more ethologically relevant sensory stimuli, but this possibility is not explored by the authors. For instance, the different behavioral options may be entirely driven by the weighting of the somatosensory maps that are driven by the sensory inputs which then drive the behaviors.

It is known that the different behaviors under investigation are more reliably triggered in response to localized mechanosensory stimuli. For instance, a rapid turn to the right is reliably elicited by a localized mechanosensory stimulus on the left side of the head. Crawling forward is reliably triggered by mechanosensory inputs to the posterior. Reversals are triggered by strong mechanosensory stimuli to the head while “hunching” may be triggered by lower sensory stimuli (these ideas are embodied in Maurice Kernan’s touch scale (and the authors may want to cite Kernan)).

Given that the individual behaviors have utility in the context of localized mechanosensory stimuli, the uniform, body-wide, stimulus of the air puff and the probabilistic behavioral responses make more sense. The network is producing a random response to a weak mechanosensory stimulus that is likely to be received uniformly across the body. Alternatively, there may be stochastic variation in how the stimulus is received by an individual animal across trials and this could be reflected in the behavioral outcome.

It is logical that the network that receives this body-wide sensory input would encompass neurons that are distributed across the brain, the thoracic, and the abdominal ganglia. However, it is not logical to conclude from this observation that the seat of “decision making” for this behavior is distributed across the brain. It is especially not appropriate to extend this finding to the rules of decision making in general. These interpretations of the results are overstated.

3.) In line 648 it is stated that the air puff was delivered continuously for 38 seconds and then there was a second delivery of 10 2 second long air puffs. Then it is stated in line 652 that the analysis was performed on behavioral probabilities for “the first seconds after stimulus onset and the transition probabilities 3 seconds after stimulus onset. This sentence is unclear, as there are 11 different stimuli that were presented. Which stimulus was analyzed? The 38 second long stimulus? All of the 2 second long stimuli?

Minor Points:

Throughout the introduction, the authors refer to “the nervous system.” It would be more accurate to write “this nervous system” or “nervous systems.”

Figure 3 panel E NompC is misspelled as nomC. Both sides of scale bar in bottom left are labeled “decrease.” It appears as if the scale to the right side should be labelled increase.

Figure 5A again has both sides of the color scale labeled as “decrease.”

On page 10 line 317 please clarify the term “mechanosensory neurons.” There are many different classes of mechanosensory neurons in the larval PNS.

On page 26 line 849 please clarify the following sentence: “Implicitly, different individuals have implicit amplitude based limits to behavior.”

What is depicted in each of the panels in Figure S3? This cannot be understood from the legend or from the text.

Reviewer #3: This manuscript uses Drosophila larval behavior to address issues concerning behavioral choice, decision-making, and selection of different types of actions, such as sequential or mutually exclusive. They perform behavioral assays on how the animals move upon being faced with a puff of air, grouping them into specific actions such as hunch, bend, back-up and stop. This is done using a collection of gal-4 lines and TNT to inhibit synaptic transmission. The neurons are then matched with EM volume of the larval CNS.

My views are somewhat mixed on this paper, which may very well reflect my lack of expertise in the area. On one hand, I greatly appreciate the type of work the authors are attempting, and the work indeed will provide a wonderful foundation for future analysis. However, many of the points were either unclear or diffuse or too broad, such that it was difficult to get what the actual take home lessons from some of these experiments were. Thus, I will outline my views to what I found not so clear in the order the paper is preseted.

1. The beginning of the results was somewhat confusing, since they say they describe wildtype and those with inactivated neurons in terms of behavioral response, and refers to Figure 1A-B. But all of Figure 1 is on wild type. So a rewording would help. It would also help Figure 1 if the colors are explained in the figure itself and not just in the legends.

2. If I understand the assay properly, a group of larvae on a surface is faced with a puff of air. What is the distribution of the types of response (hunch, bend, etc), based on the position of an animal within the surface relative to the source of stimulus? For example, from the Figure 1A, the air stream hits different animals at different positions, so do these have different outputs?

3. In 2.2, there are three paragraphs (starting with line 171, 188 and 198) where they explain the exceptions. It may help to clump all these at the end (or at once), since this breaks up the main positive message.

4. In 2.4, there is the correlation between EM and light idetified neurons, where an entire paragraph describes what has been done before. A similar thing occurs in 2.5, starting with line 347. It may help to have a table in the main figure which describes what has been done before and what is new in the current manuscript. For the latter, how indeed were these done, i.e. correlate the EM neurons with their flp-out and/or sparse lines? In a similar vein, what could explain the statement (line 373) that some could not be found in the EM colume?

5. The reasoning for their last part is a bit unclear. Line 416 "...but only later. We therefore...". Since this is the final part, it may help to remind the people again.

6. In Discussion, page 15, I was a bit confused as what their main argument is, simply because it seems like every possibility is being discussed. For example, they use the terms like perceptual decision, early sensory processing, competitive interactions between different sensorimotor pathways. One can say yes, it could be involved in any one of these steps, but this is somewhat unsatisfying, because this goes almost into asking what process being observed, indeed, is NOT considered decision-making?

7. I personally found the mere description of the neurons (Figure 4 H-J) and Figure 6, although necessary, somewhat unhelpful. Perhaps an attempt could be made to make this part scientifically more relevant, e.g., presenting these in some sort of a circuit? Similarly, Figure 8 was extremely unhelpful. Yes, it describes the data, but perhaps a better subdivisions into conceptual units or pathways might help. I don't have any concrete suggestions, unfortunately, but a bit more effort here would be great for the general audience, I think.

**Have all data underlying the figures and results presented in the manuscript been provided?**

Reviewer #1: Yes

Reviewer #2: Yes

Reviewer #3: Yes

PLOS authors have the option to publish the peer review history of their article (what does this mean?). If published, this will include your full peer review and any attached files.

Reviewer #1: No

Reviewer #2: Yes: W. Daniel Tracey

Reviewer #3: No

---

## [Decision Letter · Decision Letter 1]

30 Dec 2019

Dear Dr Jovanic,

We are pleased to inform you that your manuscript entitled "Identifying neural substrates of competitive interactions and sequence transitions during mechanosensory responses in Drosophila" has been editorially accepted for publication in PLOS Genetics. Congratulations!

Yours sincerely,

Aravinthan Samuel

Guest Editor

PLOS Genetics

Gregory Barsh

Editor-in-Chief

PLOS Genetics

Comments from the reviewers (if applicable):

Reviewer's Responses to Questions

**Comments to the Authors:**

Reviewer #2: The authors have addressed my concerns and the manuscript is suitable for publication in PLoS Genetics.

**Have all data underlying the figures and results presented in the manuscript been provided?**

Reviewer #2: Yes

PLOS authors have the option to publish the peer review history of their article (what does this mean?). If published, this will include your full peer review and any attached files.

Reviewer #2: No

**Data Deposition**

http://datadryad.org/submit?journalID=pgenetics&manu=PGENETICS-D-19-01524R1

**Press Queries**

---

## [Editor Report · Acceptance letter]

5 Feb 2020

PGENETICS-D-19-01524R1 

Identifying neural substrates of competitive interactions and sequence transitions during mechanosensory responses in Drosophila 

Dear Dr Jovanic, 

We are pleased to inform you that your manuscript entitled "Identifying neural substrates of competitive interactions and sequence transitions during mechanosensory responses in Drosophila" has been formally accepted for publication in PLOS Genetics! Your manuscript is now with our production department and you will be notified of the publication date in due course.

With kind regards,

Matt Lyles

PLOS Genetics

On behalf of:
